# Improving the Shelf-Life of Fish Burgers Made with a Mix of Sea Bass and Sea Bream Meat by Bioprotective Cultures

**DOI:** 10.3390/microorganisms10091786

**Published:** 2022-09-04

**Authors:** Lucilla Iacumin, Michela Pellegrini, Alice Sist, Giulia Tabanelli, Chiara Montanari, Cristian Bernardi, Giuseppe Comi

**Affiliations:** 1Department of Agricultural, Food, Environmental and Animal Science, University of Udine, 33100 Udine, Italy; 2Department of Agricultural and Food Sciences, University of Bologna, 40127 Bologna, Italy; 3Department of Agricultural and Food Sciences, University of Bologna, 47521 Cesena, Italy; 4Department of Veterinary Medicine and Animal Sciences, University of Milan, 20122 Lodi, Italy

**Keywords:** burgers, sea bass and sea bream meat, shelf life, bioprotection, storage

## Abstract

Seafood products are one of the most perishable foods, and their shelf life is limited by enzymatic and microbial spoilage. Developing methods to extend the shelf life of fresh fish could reduce food waste in the fishery industry, retail stores, and private households. In recent decades, the application of lactic acid bacteria (LAB) as bioprotective cultures has become a promising tool. In this study, we evaluated the use of four starter cultures, previously selected for their properties as bioprotective agents, for sea bass and sea bream burgers biopreservation. Starter cultures impacted the microbial populations, biochemical parameters (pH, TVB-N), and sensory properties of fish burgers, during 10 days of storage at 4 °C and then 20 days at 8 °C in modified atmosphere packaging (MAP). Also, storage time influenced the microbial and physicochemical characteristics of all the tested samples, except for TVB-N values, which were significantly higher in the uninoculated burgers. The volatilome changed in the different treatments, and in particular, the samples supplemented with starter presented a profile that described their rapid growth and colonization, with the production of typical molecules derived from their metabolism. The addition of bioprotective cultures avoided bloating spoilage and improved the sensory parameters of the burgers. The shelf life of the fish burgers supplemented with starter cultures could be extended up to 12 days.

## 1. Introduction

Fish are considered a functional food because of their high nutritional value, characterized by presenting components, such as mineral salts, high biological value proteins, and fatty acids, principally the polyunsaturated fatty acids of the omega 6 and omega 3 groups [1,2]. All the components are strictly necessary for the human organism because they are recognized as useful for the prevention of heart and brain diseases [1,3]. Fish consumption is widespread in Italy, with different consumption amounts from one area to another. In 2021, the consumption of fish was 21 kg/family, but the gap between the different households is significant: an average consumption of 15 kg in families with members under 30 (prefamily and new families), in fact, corresponds to a consumption of over 30 kg in older couples [www.repubblica.it (accessed on 22 August 2022); www.ismeamercati.it; (accessed on 22 August 2022)]. To increase fish consumption, two strategies are necessary: continuous training in school and the transformation of processed fish into derived products. The first strategy is very important because the preferences formed in childhood normally continue into adulthood, and for this reason, at each level of school, meal programs must include fish because they can contribute to the formation of healthy food habits [1,4].

The second strategy is more important than the first. It is well known that people sometimes give up eating fish due to the presence of fish bones and the characteristic fish odor. Therefore, fish derivatives such as burgers can be a real strategy to increase fish consumption. Indeed, burger production allows the elimination of fish bones and a decrease in the characteristic odor of fish, which are the main barriers to fish consumption [5], while maintaining the same high nutritional value of the whole fish [5,6,7]. Different works have demonstrated a positive relationship between appearance and positive hedonic perception by consumers with respect to fish derivatives [4,8].

Fish are a very highly perishable product with a shelf life (generally a few days) limited by microbial growth [9,10]. Consequently, fish burgers also have a limited shelf life. Microorganisms represent the main cause of spoilage, resulting in the formation of nitrogen compounds, sulfides, alcohols, aldehydes, ketones, and organic acids with unpleasant and unacceptable off-flavors [9,10,11,12,13].

The short shelf life of fresh seafood is often because of the growth of specific spoilage organisms [SSOs) [11,13,14]. The SSO consortium can differ among products (e.g., whole, gutted, and filleted fish) due to a series of factors, such as the composition of the initial microbiota (including the level and type of contamination), type of product, storage conditions, and microbial interactions [15]. Such differences can lead to different shelf lives of the products, even when they are stored under the same storage conditions, since different bacterial genera, species, or strains can present different growth rates or metabolism [16]. Consequently, SSO inhibition by different strategies can improve the microbial quality and prolong the shelf life of either fish or fish products. Considering the perishability of fish meat, usually to achieve lower rates of spoilage and extend the shelf life for a long time, mild procedures rather than more drastic means inhibiting SSOs represent useful approaches [10,13,16]. Among them, modified atmosphere packaging (MAP) or under vacuum, natural preservatives, essential oils (EOs), and, more recently, bioprotective cultures are obtaining success in food research activity as natural compounds with appreciable antimicrobial properties [9,10,12,13]. MAP technology implies the use of several combinations of oxygen, carbon dioxide, and nitrogen, which have different effects on the shelf life of packaged fish [17,18,19]. Again, the effectiveness of under vacuum packaging depends on the products, the storage temperature, and the experiment [9,12]. Data are often in conflict, and it is difficult to establish which technology between MAP and under vacuum is better [9,20]. However, MAP is not always sufficient to preserve processed food and requires combination with other preservation strategies, which are proposed in the literature for seafood products [21,22,23]. The most widely used approach for fresh fish burgers is based on the adoption of natural compounds that are properly encapsulated or combined with modified atmospheric conditions [22,24] or enclosed in edible films [25]. Among them, EOs exert appreciable antimicrobial properties due to the high content of phenolic derivatives, and they are potentially able to extend the shelf life of seafood [10,26].

The use of natural microbiota and/or their antimicrobial products as a biopreservation method is a recent and interesting approach to improve microbial food quality and safety [12,27]. Selected LAB strains could be used as bioprotective cultures, as they exert an antagonistic effect against potential pathogens and other undesired microorganisms [27]. By competition for nutrients, pH lowering, and the production of inhibitory compounds such as lactic acid, diacetyl, fatty acids, CO_2_, peroxide, and bacteriocins [28].

The antagonistic effect against spoilage or pathogenic microorganisms is obtained either by directly adding living cultures or purified antagonistic substances or fermentation products [9,12,28,29].

The aim of the present study was to develop fish burgers made with a mix of sea bass and sea bream meat and to improve their shelf life by bioprotective cultures. 

## 2. Material and Methods

### 2.1. Bioprotective Starter Suspension

The commercial starters (BOX-57—*Carnobacterium divergens*, *C. maltoaromaticum* and *L. sakei*; LAK-23—*L. sakei*; and FP-50—*C. divergens*, *C. maltoaromaticum*) were freeze-dried in a foil pouch. F-106 (*Lacticaseibacillus casei*) was derived from the Collection of the Department of Agricultural Food, Environmental, and Animal Sciences of the University of Udine and was grown in de Man, Rogosa and Sharpe (MRS) broth (Oxoid, Italy). After its growth, the strain was harvested by centrifugation at 8000× *g* for 10 min and then diluted in peptone water. The commercial starters were thawed, homogenized, and diluted in sterile peptone water (NaCl 0.6%, peptone, Oxoid, 0.1%, and distilled water 1 L) at the time of use. To evaluate the load of each starter, dilutions were performed in sterile peptone water, and 0.1 mL of each dilution was inoculated in the de Man, Rogosa and Sharpe (MRS) medium (Oxoid, Italy) by the double layer method for the LAB and in TSM agar (tryptic soy medium with 5% glucose, 2% NaCl, and pH 8, Oxoid, Italy) in a jar prepared for anaerobic reaction with a gas-packing anaerobic system (BBL, Becton Dickinson, USA) for the carnobacteria. The plates were incubated at 37 °C for 48–72 h, and the grown colonies were counted. Each suspension contained, on average, approximately 11 Log CFU/g.

### 2.2. Bioprotective Starter Inoculum

Sea bass (*Dicentrarchus labrax*) and sea bream (*Sparus aurata)* weighing approximately 474–578 g and 404–440 g, respectively, were headed, gutted, and filleted. Fillets were then minced, mixed, and divided into five equal batches. Each burger was composed of 68% minced fish (50% sea bass and 50% sea bream), 20.5% potato, 5.5% water, 3% rice flour, 2% vegetable fiber, and 1% NaCl. The first batch was formed into patties and directly packaged and used as a control (CTRL). The other batches were inoculated with starter cultures at a final concentration of 10^5^ CFU/g of burgers before being formed into patties. Four batches were prepared with each commercial and F-106 starter culture. The burgers were packed in MAP, consisting of 60% N_2_ and 40% CO_2_, and placed inside rectangular trays made of PET/PE/EVOH/PE: PET (Polyethylenterephtalat/PE (Polyethylene)/EVOH (Ethylene vinyl alcohol)/PE (Polyethylene), ANTIFOG—EVOH (Ethylene vinyl alcohol). The trays were laminated with a top film consisting of APET/PE/EVOH/PE: Amorphous Polyethylenterephtalat/PE (Polyethylene)/EVOH (Ethylene vinyl alcohol)/PE (Polyethylene).

The packaged burgers were stored at 4 ± 2 °C for 10 days and then at 8 ± 2 °C for 20 days, according to the challenge test proposed by AFNOR NF V01-003, 2004: hygiene and safety of foodstuffs. Guidelines for the design of an ageing test protocol for the validation of a microbiological lifetime, which reports for chilled perishable goods that in case where the cold chain is not sufficiently guaranteed, two temperatures must be used: 1/3 of the shelf life at T1 (4 °C) and 2/3 at T2 (8 °C) (abuse temperatures).

Burgers were analyzed at days 0, 6, 12, 18, 24, and 30 in triplicate to test microbial growth, physico-chemical characteristics, and sensory analyses.

### 2.3. Microbiological Analyses

Samples were analyzed by traditional microbiological methods to determine the main microbial population. Fifty grams of each sample was transferred into a sterile stomacher bag, and 100 mL of saline-peptone water (NaCl 0.8%, bacteriological peptone, Oxoid, Milan, Italy, 0.1%, distilled water 1 L) was added and mixed for 3 min in a stomacher machine (PBI, Milan, Italy). Further decimal dilutions were made in the same solution, and the following microbiological analyses were performed in duplicate agar plates: (i) total bacterial count (CBT) in Gelysate agar (gelatin sugar-free agar, Oxoid, Milan, Italy) incubated at 30 °C for 48–72 h, (ii) lactic acid bacteria (LAB) on double layer MRS agar (Oxoid, Milan, Italy), incubated at 30 °C up to 5 days, (iii) *Carnobacterium* spp. in TSM agar incubated at 30 °C for 2 days in anaerobiosis, (iv) *Enterobacteriaceae* in Violet Red Bile Glucose agar (VRBG, Oxoid, Milan, Italy), incubated at 37 °C for 48 h; all data are expressed in log CFU/g product. *L. monocytogenes* was detected according to the ISO method [30] (briefly: 25 g product were added to 225 mL of Fraser broth (Oxoid, Italy) incubated at 30 °C for 24 h, then an aliquot of this broth was streaked on Chromocult Listeria Agar according to Ottaviani/Agosti agar (Biolife, Italy), and incubated at 37 °C for 24 h. On this, agar *L. monocytogenes* produce typical blue-green colonies surrounded by an opaque halo), and *Salmonella* spp., according to the ISO method [31] (briefly: 25 g product were added to 225 mL of Buffered Peptone Water (BPW, Oxoid, Italy) incubated 18 h at 37 °C, then 1 mL of BPW in 9 mL of Rappaport Vassiliadis broth (RVB, Oxoid, Italy) incubated at 42 °C for 18–24 h. An aliquot of RVB was streaked on Xylose Lysine Tergitol 4 agar (Oxoid, Italy) incubated at 37 °C for 24 h. On this, agar the black or center black colonies were presumptive *Salmonella*). To confirm the growth of the starters, 5 colonies were collected from deMan Rogosa Sharpe plates and TSM agar and then identified by the methods reported in Iacumin et al. [32] (briefly: from MRS and TSM agars 5 colonies per plate were isolated and previous purification were subjected to Polymerase chain reaction (PCR) and the PCR products, after purification, were sent to a commercial facility for sequencing (MWG Biotech, Ebersberg, Germany). The sequences were aligned in GenBank using the Blast program version 2.2.18).

### 2.4. Physico-Chemical Analyses

The pH was detected at 3 different points using a pH meter (Basic 20, Crison Instruments, Spain) by inserting the probe directly into the product. The water activity (A_w_) was measured with an Aqua Lab 4 TE (Decagon Devices, USA), and TVB-N (total volatile basic nitrogen) was measured according to Pearson [33]. To evaluate the oxidation stability during storage, the thiobarbituric acid–reactive substances (TBARS) were determined in triplicate [34] (briefly: the total volatile basic nitrogen (TVB-N) was estimated by boiling a mix of distilled water (50 mL) and 10 g product in presence of MgO (25 mL, 2% *w/v*). The distillate was collected in a solution of boric acid and titrated with sulfuric acid in the presence of methyl red. Data are expressed in mg Nitrogen/100 g. Th thiobarbituric acid value (TBARS) was determined directly by spectrophotometric quantification of compounds obtained by the distillation of a mix consisting of distilled water (50 mL) and fish product (10 g), acidified with hydrochloric acid (2.5 mL, 4 N) until pH 1.5. Then, 5 mL of the distillate was treated with 5 mL of a solution of thiobarbituric acid (TBA), obtained by mixing TBA (0.2883) in acetic acid (90%), and placed in boiled water for 35 min. After cooling, the solution was read at 538 nm. Three analyses were performed at each sampling point and data are expressed in nmol malonaldehyde/g.

### 2.5. Analysis of Volatile Compounds (Volatilome)

Volatile organic compounds of samples were analyzed with gas chromatography-mass spectrometry coupled with solid-phase microextraction (SPME-GC‒MS) using an Agilent Hewlett–Packard 6890 GC gas chromatograph and a 5970 MSD MS detector (Hewlett–Packard, Geneva, Switzerland) equipped with a Varian (50 m × 0.32 mm × 1.2 μm) fused silica capillary column. Samples (3 g) were placed in 10 mL sterilized vials, added to a known amount of 4-methyl-2-pentanol (Sigma‒Aldrich, Steinheim, Germany) as an internal standard, and sealed by PTFE/silicon septa. The samples were heated for 10 min at 45 °C and then a fused silica SPME fiber covered with 85 μm Carboxen/Polydimethylsiloxane (CAR/PDMS) (Supelco, Steinheim, Germany) was introduced into the headspace for 40 min. Adsorbed molecules were desorbed in the gas chromatograph for 10 min. The conditions were the same as those reported by Montanari et al. [35]. Volatile peak identification was carried out by computer matching of mass spectral data with those of compounds contained in the libraries NIST 2005 and 2011. Data reported are means of three different burgers.

### 2.6. Sensory Analyses

The sensory evaluation panel consisted of 12 nonprofessional assessors. The cooked burgers were presented on white plates at room temperature. Ten burgers of the control and of each treatment were evaluated. Assessors were asked to evaluate the following descriptors: odor (fermentation, rancid, or fishy), taste (sweet, sour, pungent, or rancid), flavor (ammonia, sweet, sour, or bitter) and appearance (slime). The 12 assessors evaluated the presence or the absence of each of the nine descriptors. The results stated for each sample is the sum of the assessors who considered the presence of the descriptor out of the total of the assessors [36,37]. Then, the final score is calculated by asking the panelists to give a general evaluation of the sensory quality of the products, within a scale from 1 (excellent) to 5 (worst).

### 2.7. Statistical Analysis

Statistical elaboration was carried out by the specific software Statistica for Windows, Version 8.0 (StatSoft, Tulsa, OK, USA). Means and standard deviations were calculated, and data were elaborated by principal component analysis (PCA) and by factorial ANOVA (two factors: starter culture and time) and Tukey’s HSD test. Significant differences among samples were calculated at *p* < 0.05.

## 3. Results

### 3.1. Microbial and Physico-Chemical Characteristics

To evaluate the capability of the different starter cultures to compete with the autochthonous microbial flora of the burgers, viable counts were performed during storage. The results of viable counts of total bacterial count (CBT), Enterobacteriaceae, and lactic acid bacteria (LAB) are reported in Table 1. In all samples, CBT increased until 12 days of storage, reaching values between 5.32 and 6.47 Log CFU/g, and then decreased; at the end, it was between 3.01 and 3.58 Log CFU/g.

At the beginning of storage, the LAB counts of the inoculated samples ranged between 5.09 and 5.68 Log CFU/g, corresponding to the added amount of starter, and the control samples had 3.42 Log CFU/g. Lactic acid bacteria counts increased rapidly during the first 12 days and reached more than 7 Log CFU/g in all the samples. Thereafter, the growth continued, and at the end of the storage, the counts in the various samples reached maxima, which ranged from 8.62 to 9.18 Log CFU/g. In all the samples, the Enterobacteriaceae population increased progressively until Day 12, after which it decreased. No significant differences in counts (*p* > 0.05) of Enterobacteriaceae between the control and inoculated samples were recorded from Day 0 to Day 18 of storage. Therefore, burgers inoculated with LAK-23 presented low population amounts compared to the other samples. Listeria monocytogenes and Salmonella spp. were not detected in any of the samples. All pH profiles demonstrated clear acidification as a function of time, both in the control and when a starter culture was added. The initial pH of the burgers ranged from 6.17 to 6.31. On Day 12 of storage, the pH values decreased in all the samples, but the acidification in the inoculated samples was higher (Table 1). In fact, the pH of the control samples was 5.56, which was significantly higher (*p* < 0.05) than that of the inoculated samples, which had pH values between 4.67 and 5.09. In contrast, at Day 18, the control, together with FP-50, had lower pH values. Subsequently, the pH values of all the samples remained practically unchanged until the end of storage (4.31–4.45). The water activity (a_w_) of the burgers at Day 0 was 0.9836. Total volatile basic nitrogen (TVB-N) is known as a product of bacterial spoilage and endogenous enzyme action, and its level is often used as an index to evaluate fish quality. The levels of these compounds, which increase with the onset of microbial spoilage, are primarily responsible for the fishy odors, which increase as spoilage proceeds (Table 1). At the onset of storage, the TVB-N value of burgers was 25.60 mg N/100 g. TVB-N increased with storage time. Significant differences (*p* < 0.05) between the samples were observed from Day 18. The increase in TVB-N values was lower in the samples inoculated with FP-50 starter culture. At the end of storage (30 days), the control samples had a higher value of TVB-N. The TVB-N content of all the samples exceeded the maximum level for acceptability for marine fish (i.e., 35 mg/100 g [38]) at Day 12. The pouches were examined daily for swelling during storage. A large portion of the uninoculated burgers (40%) started swelling at 12 days. In contrast, the inoculated burgers did not become swollen during the overall storage period (Figure 1).

Principal component analysis (PCA) (data not shown) was performed to determine the relationship between design variables (time of storage and starter cultures) and microbial and physico-chemical variables. The samples on the same day of storage were grouped together, regardless of the starter cultures used. As expected, storage had a significant effect on the microbial population and physico-chemical properties of the burgers.

The TBARS content changed slightly during storage. However, either at time 0 or at 30 days of storage, no significant differences were observed in any of the samples regardless of the treatment (with or without starters added). At 0 days, the TBARS means were at the level of 1.2 ± 0.3 nmol malondialdehyde/g in all the burgers, and then they slightly increased, reaching acceptable levels at 30 days (2.1± 0.5 nmol malondialdehyde/g).

### 3.2. Volatile Compound Characteristics (VOCS)

Table 2 provides the data concerning the volatile molecules detected. The compounds are grouped according to their chemical structure, into aldehydes, ketones, alcohols, acids, and esters. The results are expressed as the ratio between the peak area of each molecule and the peak area of the standard (4-methyl-2-pentanol).

Alcohols, and in particular 1-propanol, isopropyl alcohol, and 1-penten-3-ol, were detected in higher amounts in the control at time 0. The second group was represented by aldehydes, among which hexanal, nonanal, and pentanal were the major contributors. All these aliphatic aldehydes derive mainly from lipid oxidation; they are responsible for “green notes”, but at higher concentrations, they can result in rancid perception [39].

After 6 days, aldehydes demonstrated a small decrease in the control and in the fish burgers containing the culture F-106, while an increase was observed with BOX-57, FP-50, and LAK-23 samples. Alcohols were characterized by relevant increases, especially in the samples containing bioprotective cultures. These latter were also responsible for high concentrations of ketones, such as diacetyl and acetoin, especially in BOX-57, FP-50, and LAK-23. The same three bioprotective cultures were characterized by a higher accumulation of acetic acid, while the culture F-106 seems to confirm its slower metabolism under the conditions considered here.

After 12 days, all the samples containing bioprotective cultures demonstrated higher amounts of aldehydes (in particular, hexanal) than the control, while ketones were quite similar, except for F-106, which presented a relevant accumulation of diacetyl and methyl isobutyl ketone. Acetic acid was extremely low in the control compared with samples supplemented with bioprotective cultures, among which BOX-57 was the maximum producer and F-106 was characterized by a lower amount of this acid, confirming a slower metabolism of this strain in fish burgers. Alcohols increased, especially in the control and in the samples supplemented with F-106 and BOX-57. In addition to ethanol, the alcohol increase in the fish burgers containing bioprotective cultures was due to hexanol (derived from hexanal reduction) and 1-octen-3-ol (which derives from the oxidative degradation of linoleic acid).

The samples analyzed at 18 days were characterized by a drastic reduction in the proportion of aldehydes, mainly due to the decrease in hexanal. Additionally, ketones markedly decreased in the samples containing bioprotective cultures. In particular, in all these samples, the major decreases were observed for methyl isobutyl ketone, diacetyl, and acetoin. In contrast, this chemical group increased in the control, in which great proportions of diacetyl, acetoin, and 2-butanone were detected. The control was also characterized by a relevant increase in alcohol concentration, in particular attributable to the high proportion of ethanol and 3-methyl-1-butanol. Among acids, acetic acid was again the most important molecule, whose amount, despite an increase in the control, remained remarkably higher in the samples containing bioprotective cultures. Ethyl acetate was present in a higher proportion in the control. Considering the deep changes in the volatilome, which greatly influenced the sensorial aspects and the reaching of a high value of TVB-N, it was suggested to stop the VOC analysis at 18 days. To highlight the modification of the volatile profile during storage, Figure 2 reports the results of a PCA based on the proportion of the volatile molecules detected in the samples. Factor 1 explains 27.43% and Factor 2 explains 23.45% of the variability. The different samples were grouped according to the sampling time, and within each sampling time, the control was separated from the samples containing bioprotective cultures, among which BOX-57 seems to present the most peculiar profile.

### 3.3. Sensory Characteristics

Table 3 provides the results of a sensory evaluation performed at Day 12 of storage. Inoculated burgers were also evaluated on Days 18, 24, and 30. Instead, control samples were evaluated only on Day 12, as the packages subsequently swelled. Sensory analysis results presented slight changes with the progress of storage. However, on Day 30, at the opening of packages, there was a strong ammonia odor, except for the burgers inoculated with LAK-23, which had the best sensory analysis scores during the overall storage period. The inoculation of LAB starter cultures improved the sensory attributes of fish burgers. Indeed, the bioprotective cultures reduced the spoiler activities, such as TVB-N production; consequently, the sensory attributes of sea bass and sea bream burgers were acceptable for up to 12 days of storage (10 days at 4 °C and 20 days at 8 °C). Finally, inoculated burgers did not present odors, flavors, or the sticky white slime indicative of deterioration. In contrast, a sticky, white slime was observed in some uninoculated burgers.

## 4. Discussion

### 4.1. Microbial and Physicochemical Parameters

Seafood are one of the most perishable foods, and their shelf life is limited by enzymatic and microbial spoilage. In fact, the fish matrix is particularly suitable for the development of different types of microorganisms. In addition, during evisceration and filleting, microorganisms from the raw material, manufacturing environment, and human manipulation may contaminate the flesh [40]. Several studies have been carried out to evaluate the effect of antimicrobials other than synthetic additives on the shelf life of fish burgers [41,42,43,44]. Essential oils and plant extracts have been found to be effective in extending the shelf life of the aforementioned products by limiting the growth of undesirable microorganisms [45,46]. However, in some cases, the use of these types of substances may negatively affect the sensory properties of the products [47]. In the case of minimally processed fish-based products, biopreservation seems to be an attractive alternative and novel method of preservation because of the restrictions for the application of other antimicrobial treatments, such as heat, and the lower consumer acceptance toward synthetic additives [48]. In this study, the effect of different LAB starter cultures on the quality characteristics and shelf life of sea bass and sea bream burgers was evaluated. The species used as starter cultures in this study are well adapted to the specific characteristics of the fish matrix since they are part of the natural microbiota of this kind of product [49,50,51]. The role of LAB in seafood is wide and complex; they may have no particular effect or may be responsible for spoilage, and in certain cases, they may exert a bioprotective effect in relation to undesirable bacteria [52]. The bioprotective potential of LAB strains against pathogens, especially *Listeria monocytogenes*, in different seafood products has been extensively studied. The growth of this pathogen was effectively limited by *Latilactobacillus sakei* in cold-smoked sea bass [9], sea bream [53], and cold-smoked salmon (CSS) [54], *Carnobacterium* spp. in CSS [55], *Leuconostoc gelidum* and *Lactococcus piscium* in cooked and peeled shrimp and CSS [56], just to mention a few. However, the use of LAB for improving the microbiological quality of seafood is probably more insidious than limiting the development of a pathogen, because spoilage is the result of a complex ecosystem composed of different microorganisms. The combination of starter cultures, as occurred in this study, could be useful to exert a wider antimicrobial spectrum. The antibacterial activity of organic acids produced by LAB and their ability to decrease the pH constitute one of the main mechanisms for biopreservation in food products. Despite the inocula performed, the pH reduction in the fish burgers was similar in all the samples. This could be because, in this study, the LAB populations of the control samples reached similar counts to those of the inoculated burgers. In fact, as expected, the LAB population increased during storage since modified atmosphere packaging (MAP) benefits their development, suggesting their suitability to be used to prolong the shelf life of fresh fish burgers. It is generally recognized that the absence of O_2_ and the presence of CO_2_ limit or inhibit the growth of gram-negative bacteria such as *Pseudomonas* and *Shewanella*, considered the most important spoilage bacteria in the majority of seafood [57,58]. However, *Enterobacteriaceae* can develop under these conditions and may play an important role in seafood spoilage [12]. Therefore, controlling the growth of these bacteria will be very effective in shelf-life extension. In contrast, LAB are tolerant to CO_2_ and are therefore often found as the dominant organisms in MAP fishery products [59]. Regarding the different microbial population amounts of the burgers, no strong differences were observed between the control and the inoculated samples. These results are in accordance with those of other studies. Brillet et al. [60] observed a slight inhibition of *Enterobacteriaceae*, yeasts, and endogenous LAB by the inoculation of *C. divergens* V41 in cold-smoked salmon (CSS) only when the initial natural microbial population was lower than 20 CFU/g. In fact, when it was more than 10^4^–10^5^ CFU/g, as in this study, no effect on the microbial population and physicochemical characteristics was observed. The similar counts of *Enterobacteriaceae* detected in the samples could also be related to the similar reduction in pH that occurred in all the samples, as previously mentioned. In fact, one of the main inhibitory mechanisms of *Enterobacteriaceae* by LAB is linked to a decrease in pH [61]. Another reason may be that this group of undesired microorganisms is not sensitive to the antimicrobial substances produced by the starter cultures. It is well known that most LAB bacteriocins are not active against gram-negative bacteria because of their complex outer membrane system [62]. Although the starter cultures did not have any effect on the microbial counts, they could have influenced the bacterial community species composition and diversity. In fact, it is common knowledge that the successful inoculation and implantation of starter cultures in a food product limits the growth of endogenous microorganisms and reduces bacterial diversity [63]. Most likely, in this study, the starter cultures added to the burgers, which had homofermentative metabolism, prevailed over the indigenous LAB and restricted the species variability, limiting the development of heterofermentative LAB, which caused gas formation under the packaging of the uninoculated burgers. In accordance with these results, Cao et al. [64] observed an influence of the inoculated starter *Lactiplantibacillus plantarum* 1.19 on the composition of Gram-positive bacteria of fresh tilapia fillets, but there was no effect on the Gram-negative bacteria. Heterofermentative LAB have been associated with bloating spoilage by the production of CO_2_ in several food products. Different species of the genus *Leuconostoc* were found to be responsible for bloating spoilage in vacuum-packed cooked meat and herring filets [65,66]. Members of the genus *Weissella* have been related to bulging of broiler or cooked meat packages [67]. In this study, potatoes, added to the fish batter, may have constituted an additional or main fermentable substrate for LAB since the fish matrix was characterized by a low percentage of carbohydrates (0.2–1.5% depending on the species) [40]. This may have contributed to the significant lowering of pH that occurred in all the samples and the development of CO_2_ observed in the control. The initial level of TVB-N in the fish burgers was relatively higher than that in similar products reported in other studies [43,68]. Moreover, as stated above, at the end of the storage (30 days), the TVB-N values of all the samples considerably exceeded the threshold limit suggested in the abovementioned study, which was 40 mg N/100 g product, although it is dependent on fish species and product type (e.g., whole fish, filets, slaughter method, etc.). At the end of the storage (30 days), the amount detected in the control was significantly higher than that in the inoculated burgers. TVB-N is known as a product of bacterial spoilage and endogenous enzyme action, and its level is often used as an index to evaluate fish quality [69]. The differences between the control and inoculated burgers could not be related to *Enterobacteriaceae* counts, as reported in other studies [70,71]. According to the TVB-N values, sea bass and sea bream burgers supplemented with starter were acceptable for up to 12 days of storage (10 days at 4 °C and 2 days at 8 °C). Indeed, at this time, in burgers supplemented with starter cultures, the level of TVB-N was less than or equal to 40 mg N/100 g.

The medium level of TBARS was 2.1 ± 0.5 nmol malondialdehyde/g at 30 days, independent of the treatments; consequently, these values must be accepted. According to several authors [72], food products are not rancid when TBARS values are < 8 nmol/g of the sample, slightly rancid when TBARS is between 9–20 nmol/g, and rancid and unacceptable when TBARS is > 21 nmol/g.

### 4.2. Volatilome

The presence of bioprotective starter cultures and the length of storage changed the volatile profile of the fish burger when compared with the control. The most important accumulated molecules can be derived from the pyruvate metabolism of LAB. When fermentable sugars are scarce or absent, LAB shift to mixed acid fermentation, through which, starting from pyruvate, they can accumulate additional ATP following two different pathways: the pyruvate formate lyase and the pyruvate oxidase, with the production of acetate and ethanol as end products. In addition, the species used in these trials are facultatively heterofermentative and can ferment pentose sugars present; for example, in nucleotides, producing lactic, and acetic acid [73,74]. In addition, LAB can also obtain pyruvate from other routes, such as citric acid, and amino acids such as serine [75,76]. The presence of pyruvate is also essential to produce diacetyl and acetoin, which have an important effect on the aroma profile. Other compounds that can be derived from the same precursor are acetone, 2-butanone (and its related alcohol 2-butanol), and even butyric acid [77,78]. Many other compounds detected are the result of amino acid bacterial metabolism, such as 3-methyl-1-butanol and 3-methyl butanoic acid (derived from leucine metabolism). Interestingly, 3-methyl-3-buten-1-ol and 3-methyl-2-buten-1-ol accumulated more in the presence of bioprotective cultures.

According to the results of PCA data, important differences in the volatilome were induced by the presence of bioprotective cultures. The samples supplemented with BOX-57, FP-50, and LAK-23 presented a profile that described the rapid growth and colonization of fish burgers by LAB, with the production of typical molecules derived from their metabolism. The strain *Lacticaseibacillus casei* (F-106) demonstrated slower kinetics, indicating a possible minor adaptation to the environmental conditions, particularly temperature (4–8 °C). The control presented a clearly different profile. Even if the most important molecules were the same, the kinetics of accumulation were delayed, and in contrast to other samples, after 18 days, ethanol was present in a higher proportion than acetate. These differences indicate that the LAB populations that colonized the burgers in the absence of bioprotective cultures were characterized by a different activation of metabolic routes during storage at refrigeration temperature.

### 4.3. Sensory Aspect

The growth of the starter cultures could have had an inhibitory effect on the spoilage activity of the endogenous microorganisms. It was demonstrated that some LAB have the potential to suppress the metabolic activity of certain bacteria, limiting their spoilage potential [79]. This type of interaction could be more important than the inhibition of growth, as found in the study carried out by Vasilopoulos et al. [80], in which carnobacteria limited the growth of *Brochothrix thermosphacta* without suppressing its metabolites. The panellists preferred the burgers inoculated with the starter cultures LAK-23 (*L. sakei*) and F-106 (*L. casei*). These results demonstrated that the inoculated starter cultures improved the sensory attributes of the fish burgers. Similar results were also reported by Comi et al. [81], in which the use of bioprotective cultures enhanced the microbial quality and sensory properties of meat hamburgers. To fulfill industrial requirements, bioprotective cultures should not adversely affect the sensory attributes of food products.

## 5. Conclusions

The results of this study indicated that the shelf life of sea bream and sea bass burgers, as determined by the sensory scores and physico-chemical and microbiological data, was 12 days. The inoculation of different starter cultures did not have a particular effect on the microbial populations and physico-chemical characteristics of the burgers.

The volatilome changed in the different treatments, and in particular, important differences were induced by the presence of bioprotective cultures. The samples supplemented with BOX-57, FP-50, and LAK-23 presented a profile that described the rapid growth and colonization of fish burgers by LAB, with the production of typical molecules derived from their metabolism.

The sensory attributes of burgers were affected by the presence of the bioprotective cultures, as the odors, flavors, and sticky, white slime indicative of deterioration were not observed. Additionally, inoculated burgers did not demonstrate bloating spoilage. The bioprotective cultures evaluated in this study can potentially extend the shelf life and improve the sensory properties of fish burgers, contributing to the reduction of food waste in the fish supply chain.

## Figures and Tables

**Figure 1 microorganisms-10-01786-f001:**
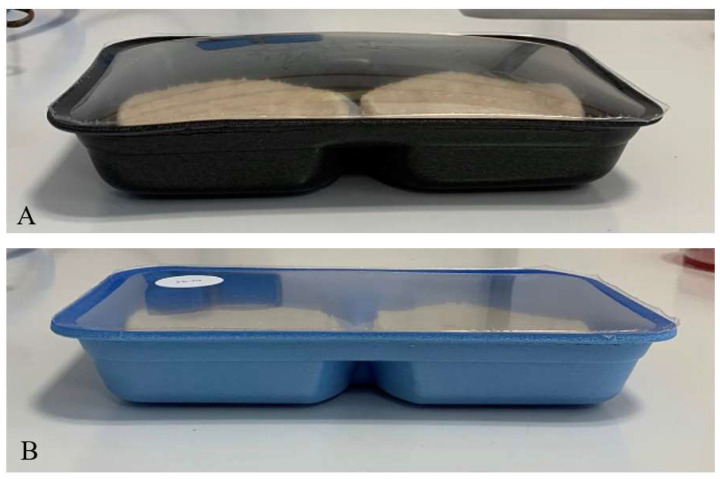
Pictures of the uninoculated (**A**) and inoculated (**B**) fish burgers at 12 days of storage.

**Figure 2 microorganisms-10-01786-f002:**
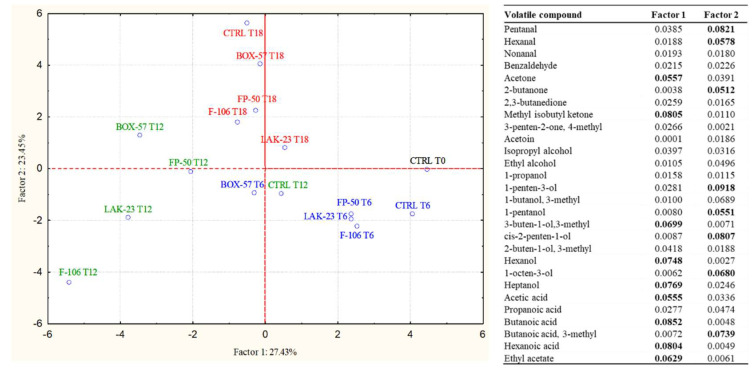
PCA case coordinates for the first two factors, explaining the influence of the different bioprotective cultures on the volatile profiles of fish burgers during storage (0, 6, 12, and 18 days). The factor contribution of the aroma compounds to Factor 1 and Factor 2 are reported as a table.

**Table 1 microorganisms-10-01786-t001:** Fate of physico-chemical and microbial characteristics of hamburgers made with a mix of sea bass and sea bream meat.

	Days						
		0	6	12	18	24	30
	Starter	Mean ± SD	Mean ± SD	Mean ± SD	Mean ± SD	Mean ± SD	Mean ± SD
**Total bacterial count** **(Log CFU/g)**	CTRL	4.76 ± 0.12 ^a^	4.89 ± 0.47 ^a^	6.47 ± 0.63 ^b^	3.58 ± 0.17 ^a^	3.68 ± 0.67 ^a^	3.26 ± 0.19 ^a^
	LAK-23	4.71 ± 0.08 ^a^	5.00 ± 0.59 ^a^	6.03 ± 0.37 ^ab^	4.06 ± 0.77 ^a^	3.20 ± 0.36 ^a^	3.49 ± 0.20 ^a^
	F-106	4.89 ± 0.09 ^a^	4.73 ± 0.06 ^a^	6.33 ± 0.13 ^b^	4.33 ± 0.49 ^a^	3.94 ± 0.44 ^a^	3.58 ± 0.07 ^a^
	FP-50	4.85 ± 0.09 ^a^	5.19 ± 0.44 ^a^	5.32 ± 0.20 ^a^	4.00 ± 0.67 ^a^	3.01 ± 0.05 ^a^	3.27 ± 0.43 ^a^
	BOX-57	5.22 ± 0.20 ^b^	5.80 ± 0.18 ^a^	5.70 ± 0.17 ^ab^	3.78 ± 0.30 ^a^	3.00 ± 0.17 ^a^	3.01 ± 0.15 ^a^
**Lactic acid bacteria** **(Log CFU/g)**	CTRL	3.42 ± 0.14 ^a^	4.44 ± 1.34 ^a^	7.55 ± 0.16 ^a^	8.73 ± 1.15 ^a^	7.83 ± 0.22 ^a^	9.18 ± 0.11 ^b^
	LAK-23	5.13 ± 0.16 ^b^	6.94 ± 0.90 ^b^	8.79 ± 0.07 ^b^	8.58 ± 0.36 ^a^	9.15 ± 0.37 ^b^	9.07 ± 0.04 ^b^
	F-106	5.09 ± 0.09 ^b^	5.90 ± 0.89 ^ab^	8.17 ± 0.15 ^ab^	8.98 ± 0.21 ^a^	8.38 ± 0.23 ^ab^	8.62 ± 0.08 ^a^
	FP-50	5.68 ± 0.15 ^c^	5.64 ± 0.27 ^ab^	7.98 ± 0.66 ^ab^	9.48 ± 0.79 ^a^	8.76 ± 0.67 ^ab^	9.11 ± 0.16 ^b^
	BOX-57	5.39 ± 0.08 ^bc^	6.35 ± 0.86 ^ab^	8.36 ± 0.22 ^ab^	9.01 ± 0.17 ^a^	8.77 ± 0.26 ^ab^	8.99 ± 0.16 ^b^
**Enterobacteriaceae** **(Log CFU/g)**	CTRL	2.74 ± 0.14 ^a^	4.67 ± 0.27 ^a^	4.58 ± 1.02 ^a^	4.01 ± 0.27 ^a^	2.39 ± 0.41 ^bc^	1.44 ± 0.36 ^b^
	LAK-23	2.83 ± 0.18 ^a^	4.40 ± 0.71 ^a^	5.45 ± 0.32 ^a^	3.55 ± 0.47 ^a^	0.52 ± 0.15 ^a^	0.48 ± 0.01 ^a^
	F-106	2.58 ± 0.12 ^a^	3.55 ± 1.55 ^a^	5.26 ± 1.13 ^a^	3.62 ± 0.32 ^a^	2.34 ± 0.38 ^bc^	3.00 ± 0.30 ^c^
	FP-50	2.63 ± 0.29 ^a^	4.31 ± 0.22 ^a^	3.97 ± 0.11 ^a^	4.08 ± 0.59 ^a^	1.48 ± 0.00 ^b^	2.40 ± 0.09 ^c^
	BOX-57	2.75 ± 0.06 ^a^	4.48 ± 0.27 ^a^	4.27 ± 0.77 ^a^	3.66 ± 0.37 ^a^	2.61 ± 0.62 ^c^	1.20 ± 0.22 ^b^
**pH**	CTRL	6.23 ± 0.03 ^ab^	6.25 ± 0.06 ^a^	5.56 ± 0.34 ^b^	4.36 ± 0.03 ^a^	4.30 ± 0.04 ^a^	4.31 ± 0.00 ^a^
	LAK-23	6.31 ± 0.03 ^b^	6.30 ± 0.11 ^a^	4.89 ± 0.01 ^a^	4.48 ± 0.02 ^b^	4.32 ± 0.08 ^a^	4.34 ± 0.07 ^ab^
	F-106	6.27 ± 0.03 ^ab^	6.31 ± 0.06 ^a^	5.09 ± 0.14 ^a^	4.52 ± 0.09 ^b^	4.39 ± 0.05 ^a^	4.38 ± 0.02 ^ab^
	FP-50	6.29 ± 0.05 ^b^	5.99 ± 0.15 ^a^	4.67 ± 0.01 ^a^	4.36 ± 0.01 ^a^	4.23 ± 0.06 ^a^	4.37 ± 0.06 ^ab^
	BOX-57	6.17 ± 0.04 ^a^	6.00 ± 0.19 ^a^	4.83 ± 0.05 ^a^	4.57 ± 0.02 ^b^	4.40 ± 0.07 ^a^	4.45 ± 0.05 ^b^
**TVB-N** **(mg N/100 g)**	CTRL	25.60 ± 3.33 ^a^	32.80 ± 2.75 ^a^	40.03 ± 1.46 ^a^	47.27 ± 0.80 ^b^	60.93 ± 1.60 ^bc^	88.63 ± 0.96 ^d^
	LAK-23	25.60 ± 3.33 ^a^	31.33 ± 2.91 ^a^	38.87 ± 1.70 ^a^	45.23 ± 2.58 ^ab^	55.13 ± 1.10 ^a^	74.37 ± 0.81 ^b^
	F-106	25.60 ± 3.33 ^a^	33.00 ± 2.95 ^a^	40.40 ± 1.13 ^a^	47.80 ± 0.60 ^b^	63.50 ± 2.69 ^c^	69.07 ± 1.25 ^a^
	FP-50	25.60 ± 3.33 ^a^	31.40 ± 0.96 ^a^	39.73 ± 1.39 ^a^	42.50 ± 1.32 ^a^	54.63 ± 2.01 ^a^	70.50 ± 1.11 ^a^
	BOX-57	25.60 ± 3.33 ^a^	32.73 ± 3.49 ^a^	39.43 ± 0.61 ^a^	45.63 ± 0.68 ^ab^	56.37 ± 1.59 ^ab^	80.53 ± 0.70 ^c^

Legend: CRT, control non inoculated; Starter added (LAK-23), *Lactobacillus sakei* bacteriocin producer; F-106, *Lacticaseibacillus casei*; FP-50, *Carnobacterium divergens, C. maltoaromaticum*; Box 57 Carnobacterium divergens, *C. maltoaromaticum, L. sakei* bacteriocin producer. Data mean ± standard deviation; mean with different letters within each day and each character (following the columns) are significantly different (*p* < 0.05).

**Table 2 microorganisms-10-01786-t002:** Mean values of volatile compounds identified at 0, 6, 12, and 18 days of storage in the different burgers. CRTL, control non inoculated; Starter added: F-106, *Lacticaseibacillus casei*; BOX-57, *Carnobacterium divergens, C. maltoaromaticum,* and *L. sakei* bacteriocin producer; FP-50, *Carnobacterium divergens, C. maltoaromaticum;* LAK-23, *Lactobacillus sakei* bacteriocin producer. Data are the mean of three replicates and are expressed as a ratio between peak area of each molecule and peak area of the internal standard (4-methyl-2-pentanol). For each molecule, significant differences according to ANOVA between samples, collected at each storage time, are indicated by the presence of different letters.

Volatile Compounds	0 Days	6 Days					12 Days					18 Days				
	CTRL	CTRL	F-106	BOX-57	FP-50	LAK-23	CTRL	F-106	BOX-57	FP-50	LAK-23	CTRL	F-106	BOX-57	FP-50	LAK-23
Pentanal	1.30	1.87 ^a^	1.76 ^a^	2.29 ^b^	1.91 ^a^	2.31 ^b^	1.52 ^a^	3.97 ^b^	2.69 ^c^	2.11 ^c^	2.65 ^c^	0.48 ^a^	1.06 ^b^	1.42 ^c^	1.42 ^c^	1.26 ^b^
Hexanal	7.28	2.48 ^a^	1.97 ^a^	8.89 ^b^	7.09 ^c^	9.52 ^b^	4.36 ^a^	10.85 ^b^	5.82 ^c^	9.52 ^b^	12.87 ^b^	2.16 ^a^	0.43 ^b^	1.02 ^c^	2.15 ^a^	4.73 ^d^
Nonanal	2.09	2.12 ^a^	1.73 ^b^	2.24 ^a^	1.22 ^c^	1.83 ^ab^	1.95 ^a^	2.43 ^b^	2.31 ^ab^	2.17 ^a^	2.47 ^b^	1.98 ^a^	2.38 ^b^	1.07 ^c^	1.00 ^c^	1.23 ^c^
Benzaldehyde	0.93	2.31 ^a^	1.97 ^a^	3.47 ^b^	2.38 ^a^	2.69 ^ac^	1.76 ^a^	3.04 ^b^	3.00 ^b^	2.13 ^a^	2.06 ^a^	1.57 ^a^	2.50 ^b^	1.95 ^a^	1.44 ^a^	2.42 ^b^
**ALDEHYDES**	**11.60**	**8.79 ^a^**	**7.42 ^a^**	**16.90 ^b^**	**12.59 ^c^**	**16.35 ^b^**	**9.59 ^a^**	**20.29 ^b^**	**13.82 ^c^**	**15.93 ^d^**	**20.05 ^b^**	**6.20 ^a^**	**6.37 ^a^**	**5.46 ^ab^**	**6.01 ^a^**	**9.64 ^c^**
Acetone	2.46	2.35	2.89	2.48	2.18	2.63	3.46 ^a^	5.29 ^b^	3.51 ^a^	3.00 ^a^	4.31 ^c^	1.20 ^a^	2.63 ^b^	2.94 ^b^	3.30 ^c^	3.63 ^c^
2-butanone	0.84	0.41 ^a^	1.09 ^b^	1.07 ^b^	0.68 ^a^	0.75 ^a^	1.97 ^a^	2.58 ^b^	1.72 ^a^	1.47 ^a^	1.88 ^a^	16.68 ^a^	1.38 ^b^	1.50 ^b^	1.59 ^b^	1.60 ^b^
Diacetyl	0.00	0.00 ^a^	0.28 ^b^	4.97 ^c^	6.91 ^d^	2.54 ^e^	1.93 ^a^	6.24 ^b^	3.22 ^c^	2.00 ^a^	3.92 ^c^	4.21 ^a^	0.30 ^b^	0.00 ^b^	0.00 ^b^	0.90 ^c^
Methyl isobutyl ketone	3.86	3.18 ^a^	4.77 ^b^	3.53 ^a^	4.51 ^b^	4.01 ^bc^	12.81 ^a^	19.30 ^b^	8.84 ^c^	11.20 ^d^	10.76 ^a^	6.36 ^a^	6.38 ^a^	6.74 ^a^	7.58 ^b^	7.34 ^b^
4-methyl,3-penten-2-one	1.74	1.60 ^a^	1.36 ^ab^	1.63 ^a^	1.80 ^a^	2.08 ^a^	0.78 ^a^	1.12 ^a^	1.66 ^b^	1.12 ^a^	0.91 ^a^	1.04 ^a^	1.21 ^a^	1.34 ^a^	1.00 ^a^	2.43 ^b^
Acetoin	0.00	0.00 ^a^	1.66 ^b^	12.18 ^c^	16.99 ^d^	8.03 ^e^	0.00 ^a^	3.82 ^b^	4.11 ^c^	2.88 ^d^	2.79 ^d^	30.57 ^a^	1.12 ^b^	1.97 ^c^	1.05 ^b^	1.89 ^c^
**KETONES**	**8.90**	**7.55 ^a^**	**12.05 ^b^**	**25.86 ^c^**	**33.07 ^d^**	**20.05 ^e^**	**20.95 ^a^**	**38.34 ^b^**	**23.06 ^ac^**	**21.68 ^a^**	**24.57 ^c^**	**60.06 ^a^**	**13.01 ^b^**	**14.49 ^c^**	**14.51 ^c^**	**17.78 ^e^**
Isopropyl alcohol	3.19	2.25 ^a^	4.53 ^b^	2.39 ^a^	3.70 ^c^	2.61 ^a^	5.27 ^a^	6.04 ^b^	3.75 ^c^	3.62 ^c^	5.33 ^a^	3.00 ^a^	4.49 ^b^	2.25 ^c^	3.43 ^a^	3.15 ^a^
Ethyl alcohol	0.57	13.68 ^a^	12.87 ^a^	24.33 ^b^	15.42 ^c^	5.90 ^d^	20.22 ^a^	20.16 ^a^	27.05 ^b^	16.48 ^c^	6.21 ^d^	62.36 ^a^	15.37 ^b^	20.74 ^c^	12.87 ^d^	9.75 ^e^
1-propanol	10.92	9.13 ^a^	10.16 ^a^	8.45 ^b^	8.84 ^ab^	10.59 ^ac^	14.40 ^a^	11.90 ^b^	10.92 ^b^	9.31 ^c^	16.19 ^d^	11.17 ^a^	8.10 ^b^	8.37 ^b^	9.56 ^c^	10.97 ^a^
1-penten-3-ol	2.90	3.78 ^a^	5.37 ^b^	4.14 ^a^	4.49 ^c^	4.45 ^c^	4.39 ^a^	7.54 ^b^	3.74 ^c^	4.24 ^a^	5.23 ^d^	3.36 ^a^	3.69 ^a^	2.55 ^b^	3.32 ^a^	3.71 ^a^
1-butanol, 3-methyl	0.00	0.00 ^a^	0.00 ^a^	2.91 ^b^	0.83 ^c^	0.63 ^c^	1.25 ^a^	1.35 ^a^	3.39 ^b^	1.61 ^a^	1.42 ^a^	11.44 ^a^	0.91 ^b^	3.07 ^c^	1.14 ^b^	0.72 ^b^
1-pentanol	1.28	3.65 ^a^	3.69 ^a^	4.67 ^b^	3.24 ^a^	2.89 ^c^	3.05 ^a^	3.60 ^b^	2.52 ^c^	3.76 ^b^	4.05 ^d^	2.45 ^a^	1.84 ^b^	1.80 ^b^	2.60 ^a^	2.72 ^a^
3-buten-1-ol,3-methyl	0.00	0.00 ^a^	0.00 ^a^	0.53 ^b^	0.17 ^ab^	0.00 ^ab^	0.33 ^a^	0.72 ^b^	0.54 ^a^	0.93 ^b^	0.88 ^b^	0.25 ^a^	0.67 ^b^	0.70 ^b^	0.96 ^b^	0.91 ^b^
cis 2-penten-1-ol	1.71	3.26 ^a^	3.63 ^a^	3.17 ^a^	3.15 ^a^	2.87 ^b^	1.79 ^a^	3.03 ^b^	1.64 ^a^	2.27 ^c^	1.78 ^a^	1.66 ^a^	1.85 ^a^	1.13 ^b^	1.50 ^a^	1.62 ^a^
2-buten-1-ol, 3-methyl	0.00	0.00 ^a^	0.00 ^a^	0.54 ^b^	0.25 ^b^	0.69 ^b^	1.15 ^a^	2.21 ^b^	4.00 ^c^	3.06 ^d^	2.98 ^d^	0.00 ^a^	5.80 ^b^	4.19 ^c^	4.23 ^c^	3.28 ^d^
Hexanol	1.94	2.10 ^a^	2.81 ^b^	5.39 ^c^	3.61 ^d^	4.11 ^d^	2.47 ^a^	5.10 ^b^	6.19 ^c^	4.46 ^d^	4.36 ^d^	4.70 ^a^	3.41 ^b^	4.24 ^b^	3.63 ^b^	3.03 ^bd^
1-octen-3-ol	2.78	5.48	5.59	5.49	5.54	5.21	3.92 ^a^	7.47 ^b^	4.21 ^ac^	4.45 ^c^	4.35 ^c^	4.51 ^a^	4.04 ^a^	2.95 ^b^	2.85 ^b^	3.05 ^b^
Heptanol	1.12	1.38 ^a^	0.39 ^b^	0.38 ^b^	1.09 ^a^	1.19 ^a^	0.34 ^a^	0.00 ^b^	0.21 ^a^	0.00 ^b^	0.00 ^b^	0.00	0.00	0.00	0.00	0.00
**ALCOHOLS**	**26.41**	**44.71 ^a^**	**49.03 ^b^**	**62.39 ^c^**	**50.34 ^d^**	**41.14 ^a^**	**58.57 ^a^**	**69.13 ^b^**	**68.17 ^b^**	**54.19 ^a^**	**52.80 ^ac^**	**104.90 ^a^**	**50.17 ^b^**	**51.98 ^b^**	**46.07 ^b^**	**42.94 ^bc^**
Acetic acid	1.12	0.71 ^a^	1.24 ^b^	15.57 ^c^	3.15 ^d^	5.32 ^e^	2.96 ^a^	24.71 ^b^	66.15 ^c^	38.58 ^d^	38.87 ^d^	16.21 ^a^	53.03 ^b^	52.55 ^b^	48.10 ^c^	39.48 ^d^
Propanoic acid	0.00	0.26 ^a^	0.00 ^b^	0.70 ^a^	0.40 ^a^	0.55 ^a^	0.46 ^a^	0.94 ^a^	1.26 ^ab^	0.95 ^a^	1.39 ^b^	3.49 ^a^	0.58 ^b^	0.75 ^b^	0.64 ^b^	0.73 ^b^
Butanoic acid	0.00	0.00 ^a^	0.00 ^a^	0.46 ^b^	0.28 ^b^	0.40 ^b^	0.31 ^a^	0.90 ^b^	1.62 ^c^	1.24 ^b^	1.46 ^bc^	0.23 ^a^	1.12 ^b^	1.03 ^b^	0.97 ^b^	0.75 ^b^
Butanoic acid, 3-methyl	0.00	0.00 ^a^	0.00 ^a^	0.74 ^b^	0.00 ^a^	0.19 ^ab^	0.00 ^a^	0.00 ^a^	0.74 ^b^	0.24 ^ab^	0.00 ^a^	0.67 ^a^	0.36 ^a^	1.26 ^b^	0.32 ^a^	0.00 ^c^
Hexanoic acid	1.11	0.61 ^a^	0.38 ^a^	1.85 ^b^	0.78 ^a^	0.97 ^a^	0.70 ^a^	2.21 ^b^	2.60 ^b^	2.19 ^b^	2.41 ^b^	0.95 ^a^	2.56 ^b^	2.16 ^b^	1.74 ^c^	1.39 ^a^
**ACIDS**	**2.23**	**1.57 ^a^**	**1.62 ^a^**	**19.32 ^b^**	**4.61 ^c^**	**7.42 ^d^**	**4.43 ^a^**	**28.75 ^b^**	**72.36 ^c^**	**43.20 ^d^**	**44.13 ^d^**	**21.56 ^a^**	**57.65 ^b^**	**57.75 ^b^**	**51.77 ^c^**	**42.35 ^d^**
Ethyl acetate	1.08	3.93 ^a^	6.83 ^b^	5.99 ^c^	4.05 ^d^	2.50 ^e^	5.81 ^a^	14.15 ^b^	10.44 ^c^	6.91 ^d^	5.35 ^a^	13.99 ^a^	5.13 ^b^	5.89 ^b^	7.00 ^c^	4.88 ^bd^
**ESTERS**	**1.08**	**3.93 ^a^**	**6.83 ^b^**	**5.99 ^c^**	**4.05 ^d^**	**2.50 ^e^**	**5.81 ^a^**	**14.15 ^b^**	**10.44 ^c^**	**6.91 ^d^**	**5.35 ^a^**	**13.99 ^a^**	**5.13 ^b^**	**5.89 ^b^**	**7.00 ^c^**	**4.88 ^bd^**

**Table 3 microorganisms-10-01786-t003:** The sensory panel scores of cooked fish burgers.

Sensory Attributes	Day 12				
	CTRL	BOX-57	F-106	FP-50	LAK-23
**Fermentation**	8/12	6/12	4/12	5/12	3/12
**Rancid**	7/12	4/12	3/12	3/12	2/12
**Sweet**	2/12	1/12	1/12	1/12	1/12
**Pungent**	7/12	2/12	3/12	3/12	2/12
**Fish**	12/12	12/12	12/12	12/12	12/12
**Sour**	4/12	5/12	5/12	4/12	3/12
**Bitter**	6/12	4/12	2/12	2/12	2/12
**Ammonia**	4/12	3/12	1/12	2/12	1/12
**Slimes**	2/12	0/12	0/12	1/12	0/12
**Final scores ^a^**	5	4	2	3	1

Legend: CTRL, no starter; BOX-57 (*C. maltaromaticum, C. divergens,* and *L. sakei*); F-106 (*Lacticaseibacillus casei*); FP-50 (*C. maltaromaticum, C. divergens*); LAK-23 (*L. sakei*). Data: sum of the assessors’ identifiers of the presence of the descriptor out of the total of the assessors; ^a^ Final scores, the assessors were requested to ranked the products within a scale from 1 (excellent) to 5 (worst).

## Data Availability

The data in this study are readily available upon reasonable request to the corresponding author.

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
