# Peer review of "Improving the Shelf-Life of Fish Burgers Made with a Mix of Sea Bass and Sea Bream Meat by Bioprotective Cultures"

_microorganisms, 2022, doi:10.3390/microorganisms10091786_

Round 1

Reviewer 1 Report

This paper need a major revision before publication, and further review are required for the revised manuscript.

1.     The title of this manuscript should be modified. The current title is a little bit long and hard to read.

2.     Line 23-24: ‘in particular’ occurred repeated in adjacent sentences. Please modify the expressions.

3.     Line 41: ‘[prefamily and new families)’, please check the punctuation.

4.     Line 45-48: This part is not very necessary for the whole manuscript. Maybe it could be abridged into 1 or 2 sentences.

5.     Line 99-101: Please re-write this sentence to make it clearer.

6.     Line 99-101: ‘11 Log CFU/mL’. Please check the unit.

7.     Table 1: It is great recommended to change the data in table to the figure format.

8.     Line 241: The PCA results could be added in the supplementary data.

9.     The title of each section should be rewrite. For example, ‘3.3. Effect on sensory properties‘ seems not clear enough.

Author Response

Response to Reviewer 1 Comments

Dear Referee

Manuscript ID: microorganisms-1884900

  

Improving the shelf-life of fish burgers made with sea bass and sea bream meat by bioprotective cultures.

Journal: Microorganisms

Answer to the referee.

The authors would like to thank the reviewers for their careful reading of the manuscript and the resulting constructive comments and suggestions. Basically, we agree with all of the points raised by the reviewers, and wherever possible the manuscript has been modified as recommended. All reviewer comments are in black plain font, whereas our response is described in red plain font.

We have made the changes and corrections on the basis of the reviewer’s suggestions. We evaluated the comments and prepared a point-by-point response to each one of them.

Comments from the editors and reviewers:

The manuscript has been modified including most of the suggestions. Then, it can be now accepted for publication.

Reviewers' comments:

Reviewer 1  

  • This paper need a major revision before publication, and further review are required for the revised manuscript.

  1. The title of this manuscript should be modified. The current title is a little bit long and hard to read.

Thanks – Answer - Lines 2-3 - I changed – Improving the shelf-life of fish burgers made with sea bass and sea bream meat by bioprotective cultures.

  1. Line 23-24: ‘in particular’ occurred repeated in adjacent sentences. Please modify the expressions.

Thanks – Answer - Lines 23-26 - I changed - The volatilome changed in the different treatments, and in particular, the samples supplemented with starter presented a profile that described their rapid growth and colonization, with the production of typical molecules derived from their metabolism.

  1. Line 41: ‘[prefamily and new families)’, please check the punctuation.

Thanks – Answer – Lines 41 -  I changed - (prefamily and new families).

  1. Line 45-48: This part is not very necessary for the whole manuscript. Maybe it could be abridged into 1 or 2 sentences.

Thanks – Answer – Lines 44-47 - I changed : The first strategy is very important because the preferences formed in childhood normally continue into adulthood and for this reason, at each level of school, meal programs must include fish food because they can contribute the formation of healthy food habits [1,4].

  1. Line 99-101: Please re-write this sentence to make it clearer.

Thanks – Answers – Lines 90-93 - I modified: Selected LAB strains could be used as bioprotective cultures, as they exert an antagonistic effect against potential pathogens and other undesired microorganisms [27]. By competition for nutrients, pH lowering and the production of inhibitory compounds such as lactic acid, diacetyl, fatty acids, CO2, peroxide, and bacteriocins [28].

  1. Line 99-101: ‘11 Log CFU/mL’. Please check the unit.

Thanks – Answer – Lines 116 -  I modified: 11 log CFU/g

  1. Table 1: It is great recommended to change the data in table to the figure format.

Thanks – Answer: I changed it, 260-265 (see the revised paper)

  1. 8.     Line 241: The PCA results could be added in the supplementary data.

Thanks – Answer – It is impossible because the other referees suggested to leave it.

  1. The title of each section should be rewrite. For example, ‘3.3. Effect on sensory properties‘ seems not clear enough.

Thanks – Answer: I changed

Lines 221 - 3.1. Effect on microbial and physicochemical parameters – Microbial and physico-chemical characteristics

Lines 277-  3.2. Effect on volatile components (VOCS) – Volatile compounds (VOCS) characteristics

Lines 355 - 3.3. Effect on sensory properties – Sensory characyeristics

Reviewer 2 Report

Pls rewrite intro and discussion section in concise manner.

Author Response

Response to Reviewer 2 Comments

Dear Referee

Manuscript ID: microorganisms-1884900

  

Improving the shelf-life of fish burgers made with sea bass and sea bream meat by bioprotective cultures.

Journal: Microorganisms

Answer to the referee.

The authors would like to thank the reviewers for their careful reading of the manuscript and the resulting constructive comments and suggestions. Basically, we agree with all of the points raised by the reviewers, and wherever possible the manuscript has been modified as recommended. All reviewer comments are in black plain font, whereas our response is described in red plain font.

We have made the changes and corrections on the basis of the reviewer’s suggestions. We evaluated the comments and prepared a point-by-point response to each one of them.

Comments from the editors and reviewers:

The manuscript has been modified including most of the suggestions. Then, it can be now accepted for publication.

Reviewers' comments:

Reviewer 2

Pls rewrite intro and discussion section in concise manner.

Thanks – Answer – I changed

  1. Introduction

Fish are considered a functional food because of their high nutritional value, characterized by presenting components, such as mineral salts, high biological value proteins and fatty acids, principally the polyunsaturated ones of the omega 6 and omega 3 groups [1,2]. All the components are strictly necessary for the human organism because they are recognized as useful for the prevention of heart and brain diseases [1,3]. Fish consumption is widespread in Italy, with different consumption amounts from one area to another. In 2021, the consumption of fish was 21 kg/family, but the gap between the different households is significant: an average consumption of 15 kg in families with members under 30 (prefamily and new families), in fact, corresponds to a consumption of over 30 kg in older couples [www.repubblica.it; www.ismeamercati.it]. To increase fish consumption, two strategies are necessary: continuous training in school and the transformation of processed fish into derived products. The first strategy is very important because the preferences formed in childhood normally continue into adulthood and for this reason, at each level of school, meal programs must include fish food because they can contribute the formation of healthy food habits [1,4].

The second strategy is more important than the first. It is well known that people sometimes give up eating fish due to the presence of fish bones and the characteristic fish odor. Therefore, fish derivatives such as burgers can be a real strategy to increase fish consumption. Indeed, burger production allows the elimination of fish bones and a decrease in the characteristic odor of fish, which are the main barriers to fish consumption [5], maintaining the same high nutritional value of the whole fish [5-7]. Different works have demonstrated a positive relationship between appearance and positive hedonic perception by consumers with respect to fish derivatives [4,8].

Fish are a very highly perishable product with a shelf life (generally a few days) limited by microbial growth [9,10]. Consequently, fish burgers also have a limited shelf life. Microorganisms represent the main cause of spoilage, resulting in the formation of nitrogen compounds, sulfides, alcohols, aldehydes, ketones, and organic acids with unpleasant and unacceptable off-flavors [9-13].

The short shelf life of fresh seafood is often because of the growth of specific spoilage organisms [SSOs) [11,13,14].

The SSO consortium can differ among products, e.g., whole, gutted and filleted fish, due to a series of factors, such as the composition of the initial microbiota (including the level and type of contamination), type of product, storage conditions, and microbial interactions [15]. Such differences can lead to different shelf lives of the products, even when they are stored under the same storage conditions, since different bacterial genera, species or strains can present different growth rates or metabolism [16]. Consequently, SSO inhibition by different strategies can improve the microbial quality and prolong the shelf life of either fish or fish products. Considering the pershability of fish meat, usually to achieve lower rates of spoilage and extend the shelf life for a long time, mild procedures rather than more drastic means inhibiting SSOs represent useful approaches [10,13,16]. Among them, modified atmosphere packaging (MAP) or under vacuum, natural preservatives, essential oils (EOs) and, more recently, bioprotective cultures are obtaining success in food research activity as natural compounds with appreciable antimicrobial properties [9,10,12,13]. MAP technology implies the use of several combinations of oxygen, carbon dioxide and nitrogen, which have different effects on the shelf life of packaged fish [17-19]. Again, the effectiveness of under vacuum packaging depends on the products, the storage temperature and the experiment [9,12]. Data are often in conflict, and it is difficult to establish which technology between MAP and Under Vacuum is better [9,20]. However, MAP is not always sufficient to preserve processed food and requires combination with other preservation strategies, which are proposed in the literature for seafood products [21-23]. The most widely used approach for fresh fish burgers is based on the adoption of natural compounds that are properly encapsulated or combined with modified atmospheric conditions [22,24] or enclosed in edible films [25]. Among them, EOs exert appreciable antimicrobial properties due to the high content of phenolic derivaties and they are potentially able to extend the shelf life of seafood [10,26].

The use of natural microbiota and/or their antimicrobial products as a biopreservation method is a recent and interesting approach to improve microbial food quality and safety [12,27]. Selected LAB strains could be used as bioprotective cultures, as they exert an antagonistic effect against potential pathogens and other undesired microorganisms [27]. By competition for nutrients, pH lowering and the production of inhibitory compounds such as lactic acid, diacetyl, fatty acids, CO2, peroxide, and bacteriocins [28].

The antagonistic effect against spoilage or pathogenic microorganisms is obtained either by directly adding living cultures or purified antagonistic substances or fermentation products [9,12,28,29].

The aim of the present study was to develop fish burgers made with a mix of sea bass and sea bream meat and to improve their shelf life by bioprotective cultures.

  1. Discussion

4.1. Microbial and physicochemical parameters

Seafood are one of the most perishable foods, and their shelf life is limited by enzymatic and microbial spoilage. In fact, the fish matrix is particularly suitable for the development of different types of microorganisms. In addition, during evisceration and filleting, microorganisms from the raw material, manufacturing environment and human manipulation may contaminate the flesh [40]. Several studies have been carried out to evaluate the effect of antimicrobials other than synthetic additives on the shelf life of fish burgers [41-44]. Essential oils and plant extracts have been found to be effective in extending the shelf life of the aforementioned products by limiting the growth of undesirable microorganisms [45,46]. However, in some cases, the use of these types of substances may negatively affect the sensory properties of the products [47]. In the case of minimally processed fish-based products, biopreservation seems to be an attractive alternative and novel method of preservation because of the restrictions for the application of other antimicrobial treatments, such as heat, and the lower consumer acceptance toward synthetic additives [48]. In this study, the effect of different LAB starter cultures on the quality characteristics and shelf life of sea bass and sea bream burgers was evaluated. The species used as starter cultures in this study are well adapted to the specific characteristics of the fish matrix since they are part of the natural microbiota of this kind of product [49-51]. The role of LAB in seafood is wide and complex, they may have no particular effect or may be responsible for spoilage, and in certain cases, they may exert a bioprotective effect in relation to undesirable bacteria [52]. The bioprotective potential of LAB strains against pathogens, especially Listeria monocytogenes, in different seafood products has been extensively studied. The growth of this pathogen was effectively limited by Latilactobacillus sakei in cold-smoked sea bass [9], sea bream [53], cold-smoked salmon (CSS) [54]; Carnobacterium spp. in CSS [55]; Leuconostoc gelidum and Lactococcus piscium in cooked and peeled shrimp and CSS [56], just to mention a few. However, the use of LAB for improving the microbiological quality of seafood is probably more insidious than limiting the development of a pathogen because spoilage is the result of a complex ecosystem composed of different microorganisms. The combination of starter cultures, as occurred in this study, could be useful to exert a wider antimicrobial spectrum. The antibacterial activity of organic acids produced by LAB and their ability to decrease the pH constitute one of the main mechanisms for biopreservation in food products. Despite the inocula performed, the pH reduction in the fish burgers was similar in all the samples. This could be because in this study, the LAB populations of the control samples reached similar counts to those of the inoculated burgers. In fact, as expected, the LAB population increased during storage since modified atmosphere packaging (MAP) benefits their development, suggesting their suitability to be used to prolong the shelf life of fresh fish burgers. It is generally recognized that the absence of O2 and the presence of CO2 limit or inhibit the growth of gram-negative bacteria such as Pseudomonas and Shewanella, considered the most important spoilage bacteria in the majority of seafood [57,58]. However, Enterobacteriaceae can develop under these conditions and may play an important role in seafood spoilage [12]. Therefore, controlling the growth of these bacteria will be very effective in shelf-life extension. In contrast, LAB are tolerant to CO2 and are therefore often found as the dominant organisms in MAP fishery products [59]. Regarding the different microbial population amounts of the burgers, no strong differences were observed between the control and the inoculated samples. These results are in accordance with those of other studies. Brillet et al. [60] observed a slight inhibition of Enterobacteriaceae, yeasts and endogenous LAB by the inoculation of C. divergens V41 in cold-smoked salmon (CSS) only when the initial natural microbial population was lower than 20 CFU/g. In fact, when it was more than 104-105 CFU/g, as in this study, no effect on the microbial population and physicochemical characteristics was observed. The similar counts of Enterobacteriaceae detected in the samples could also be related to the similar reduction in pH that occurred in all the samples, as previously mentioned. In fact, one of the main inhibitory mechanisms of Enterobacteriaceae by LAB is linked to a decrease in pH [61]. Another reason may be that this group of undesired microorganisms is not sensitive to the antimicrobial substances produced by the starter cultures. It is well known that most LAB bacteriocins are not active against gram-negative bacteria because of their complex outer membrane system [62]. Although the starter cultures did not have any effect on the microbial counts, they could have influenced the bacterial community species composition and diversity. In fact, it is common knowledge that the successful inoculation and implantation of starter cultures in a food product limits the growth of endogenous microorganisms and reduces bacterial diversity [63]. Most likely, in this study, the starter cultures added to the burgers, which had homofermentative metabolism, prevailed over the indigenous LAB and restricted the species variability, limiting the development of heterofermentative LAB that caused gas formation under packaging of the uninoculated burgers. In accordance with these results, Cao et al. [64] observed an influence of the inoculated starter Lactiplantibacillus plantarum 1.19 on the composition of Gram-positive bacteria of fresh tilapia fillets, but there was no effect on the Gram-negative bacteria. Heterofermentative LAB have been associated with bloating spoilage by the production of CO2 in several food products. Different species of the genus Leuconostoc were found to be responsible for bloating spoilage in vacuum-packed cooked meat and herring filets [65,66]. Members of the genus Weissella have been related to bulging of broiler or cooked meat packages [67]. In this study, potatoes, added to the fish batter, may have constituted an additional or main fermentable substrate for LAB since the fish matrix was characterized by a low percentage of carbohydrates (0.2-1.5% depending on the species) [40]. This may have contributed to the significant lowering of pH that occurred in all the samples and the development of CO2 observed in the control. The initial level of TVB-N in the fish burgers was relatively higher than that in similar products reported in other studies [43,68]. Moreover, as stated above, at the end of the storage (30 days), the TVB-N values of all the samples considerably exceeded the threshold limit suggested in the abovementioned study, which was 40 mg N/100 g product, although it is dependent on fish species and product type (e.g., whole fish, filets, slaughter method, etc.). At the end of the storage (30 days), the amount detected in the control was significantly higher than that in the inoculated burgers. TVB-N is known as a product of bacterial spoilage and endogenous enzyme action, and its level is often used as an index to evaluate fish quality [69]. The differences between the control and inoculated burgers could not be related to Enterobacteriaceae counts, as reported in other studies [70,71]. According to the TVB-N values, sea bass and sea bream burgers supplemented with starter were acceptable for up to 12 days of storage (10 days at 4 °C and 2 days at 8 °C). Indeed, at this time, in burgers supplemented with starter cultures, the level of TVB-N was less than or equal to 40 mg N/100 g.

Figure 1. Pictures of the uninoculated (A) and inoculated (B) fish burgers at 12 days of storage.

The medium level of TBARS was 2.1± 0.5 nmol malondialdehyde/g at 30 days independent of the treatments; consequently, these values must be accepted. According to several authors [72], food products are not rancid when TBARS values are < 8 nmol⁄g of the sample, slightly rancid when TBARS is between 9–20 nmol⁄g, and rancid and unacceptable when TBARS is > 21 nmol⁄g.

4.2. Volatilome

The presence of bioprotective starter cultures and the length of storage changed the volatile profile of the fish burger when compared with the control. The most important accumulated molecules can be derived from the pyruvate metabolism of LAB. When fermentable sugars are scarce or absent, LAB shift to mixed acid fermentation through which, starting from pyruvate, they can accumulate additional ATP, following two different pathways: the pyruvate formate lyase and the pyruvate oxidase, with the production of acetate and ethanol as end products. In addition, the species used in these trials are facultatively heterofermentative and can ferment pentose sugars present, for example, in nucleotides, producing lactic and acetic acid [73,74]. In addition, LAB can also obtain pyruvate from other routes, such as citric acid and amino acids such as serine [75,76]. The presence of pyruvate is also essential to produce diacetyl and acetoin, which have an important effect on the aroma profile. Other compounds that can be derived from the same precursor are acetone, 2-butanone (and its related alcohol 2-butanol) and even butyric acid [77,78]. Many other compounds detected are the result of amino acid bacterial metabolism, such as 3-methyl-1-butanol and 3-methyl butanoic acid (derived from leucine metabolism). Interestingly, 3-methyl-3-buten-1-ol and 3-methyl-2-buten-1-ol accumulated more in the presence of bioprotective cultures.

According to the results of PCA data, important differences in the volatilome were induced by the presence of bioprotective cultures. The samples supplemented with BOX-57, FP-50 and LAK-23 presented a profile that described the rapid growth and colonization of fish burgers by LAB, with the production of typical molecules derived from their metabolism. The strain Lacticaseibacillus casei (F-106) showed slower kinetics, indicating a possible minor adaptation to the environmental conditions, particularly temperature (4-8 °C). The control presented a clearly different profile. Even if the most important molecules were the same, the kinetics of accumulation were delayed, and in contrast to other samples, after 18 days, ethanol was present in a higher proportion than acetate. These differences indicate that the LAB populations that colonized the burgers in the absence of bioprotective cultures were characterized by a different activation of metabolic routes during storage at refrigeration temperature.

4.3. Sensory aspect

Inoculated burgers did not present odors, flavors or sticky white slime indicative of deterioration. In contrast, a sticky-white slime was observed in some uninoculated burgers. The growth of the starter cultures could have had an inhibitory effect on the spoilage activity of the endogenous microorganisms. It was demonstrated that some LAB have the potential to suppress the metabolic activity of certain bacteria, limiting their spoilage potential [79]. This type of interaction could be more important than the inhibition of growth, as seen in the study carried out by Vasilopoulos et al. [80], in which carnobacteria limited the growth of Brochothrix thermosphacta without suppressing its metabolites. The panellists preferred the burgers inoculated with the starter cultures LAK-23 (L. sakei) and F-106 (L. casei). These results showed that the inoculated starter cultures improved the sensory attributes of the fish burgers. Similar results were also reported by Comi et al. [81], in which the use of bioprotective cultures enhanced the microbial quality and sensory properties of meat hamburgers. To fulfill industrial requirements, bioprotective cultures should not adversely affect the sensory attributes of food products.

Reviewer 3 Report

This manuscript is about using lactic acid bacteria (LAB) as bioprotective cultures for the storage of fish burgers. Although the novelty of the study is limited, it can provide some information for the application of different LAB strains in food protection. 

Other comments:

All tables,especially Table 2, should be improved to make them more clear.

Author Response

Response to Reviewer 3 Comments

Dear Referee

Manuscript ID: microorganisms-1884900

  

Improving the shelf-life of fish burgers made with sea bass and sea bream meat by bioprotective cultures.

Journal: Microorganisms

Answer to the referee.

The authors would like to thank the reviewers for their careful reading of the manuscript and the resulting constructive comments and suggestions. Basically, we agree with all of the points raised by the reviewers, and wherever possible the manuscript has been modified as recommended. All reviewer comments are in black plain font, whereas our response is described in red plain font.

We have made the changes and corrections on the basis of the reviewer’s suggestions. We evaluated the comments and prepared a point-by-point response to each one of them.

Comments from the editors and reviewers:

The manuscript has been modified including most of the suggestions. Then, it can be now accepted for publication.

Reviewers' comments:

Reviewer 3

This manuscript is about using lactic acid bacteria (LAB) as bioprotective cultures for the storage of fish burgers. Although the novelty of the study is limited, it can provide some information for the application of different LAB strains in food protection.

Other comments:

All tables,especially Table 2, should be improved to make them more clear.

Thanks – Answer – I changed table 1 and 2 . I modified Table 3 adding the follow legend

Lines 371-374 - Legend: CTRL: no starter; BOX-57 (C. maltaromaticum, C. divergens, L. sakei), F-106 (Lacticaseibacillus casei), FP-50 (C. maltaromaticum, C. divergens); LAK-23 (L. sakei). Data: sum of the assessors identifiers of the presence of the descriptor out of the total of the assessors; a Final scores: the assessors requested to ranked the products within the scale from 1 (excellent) to 4 (worst).

Sorry, see the revised paper, the tables lost the formatting.

Reviewer 4 Report

This study evaluated the use of four LAB starter cultures, previously selected for their properties as bioprotective agents, for sea bass and sea bream burgers biopreservation for storage at 4 °C for 10 days and at 8°C for the remaining 20 days in modified atmosphere packaging. Microbial populations, biochemical parameters (pH, TVB-N), and sensory properties of fish burgers were analized.

The work is interesting, but the presentation of the results and the discussion should be reviewed. Below is a list of suggestions.

Title: could be shorter

Line 19-20: This part of the sentence is not understood “followed by 30 days and volatilization for 18 days ….”

Line 20-22: This sentence contradicts what was said above: “In general, microbial and physicochemical parameters were not affected by the use of starter cultures but by the storage time, except for TVB-N values, which were significantly higher in the uninoculated burgers”

Line 42: This quote cannot be found: [www.mangimialimenti.it]

Line 82: Clarify what the acronym VP means.

Line 104-107: The objective is not worded properly.

Line 136-137: Clarify what the acronyms means.

Line 137-138: Why was a change in storage conditions made to day 10?

Line 149-150: Incubation of MRS agar: up to 3 days at 35ºC or up to 5 days at 30ºC.

Line 152-154: The detection methodology for L. monocytogenes and Salmonella spp. should be briefly described.

Line 154-156: The identification methodology of MRS and TSM isolates should be briefly described.

Line 160-162: The methodology to determine TVB-N and oxidative stability should be briefly described.

Line 199-200: “Total bacterial counts were similar for all the samples 199 throughout the storage period and were not affected by starter inoculum.” On day 0 and 12 there are differences

Line 214-216: “In fact, the pH of the control samples was 5.56, which was significantly higher (p < 0.05) than that of the inoculated samples, which had pH values lower than 5.” Sample F-106 presented a pH value greater than 5

Line 263-264: “Alcohols, and in particular 1-propanol, isopropyl alcohol and 1-penten-3-ol, were detected in higher amounts in the control at time 0” Time 0 data is missing for inoculated samples in the table

Line 268- 270. “After 6 days, aldehydes showed a small decrease in the control and in the fish burgers 268 containing the culture F-106, while an increase was observed when BOX-57, FP-50 and 269 LAK-23 were added” The way in which the data is presented does not speed up the reading of the evolution of the data over time.

Line 301: “Factor 1 explains 27.43% and Factor 301 2 explains 23.45% of the variability.” The definition of factor 1 factor 2 is missing.

Line 385: “20 CFU/g”. Is Is it correct?

Line 398-340: “… the starter cultures added to the burgers, 398 which had homofermentative metabolism, prevailed over the indigenous LAB and re- 399 stricted the species variability…” Is this conclusion correct? Do the data obtained allow us to reach this statement?Table 1: Not all data corresponds to Log.

Line 401: It is not correct to quote figures in the discussion

Line 427-429: “It can be concluded that the bioprotective cultures used in this study resulted in effective elimination of bloating spoilage” Is it correct?

Line 430: The figure should be cited in the results section.

Line 461: “The strain Lacticaseibacillus casei (F-106) showed slower kinetics” Is it correct?

Line 470: “Inoculated burgers did not present odors, flavors or sticky white slime indicative of deterioration.” Is this described in the results?

Table 2: The data is outside the corresponding line. Some of the acronyms are not clarified in the legend. It does not present the SD. Nor is it discriminated which results are statistically different. This table should be redesigned to provide easy reading of the results.

Table 3: It is not understood what the proportions indicated in the table refer to and how the final score is calculated. In materials and methods it is not explained.

Author Response

Response to Reviewer 4 Comments

Dear Referee

Manuscript ID: microorganisms-1884900

  

Improving the shelf-life of fish burgers made with sea bass and sea bream meat by bioprotective cultures.

Journal: Microorganisms

Answer to the referee.

The authors would like to thank the reviewers for their careful reading of the manuscript and the resulting constructive comments and suggestions. Basically, we agree with all of the points raised by the reviewers, and wherever possible the manuscript has been modified as recommended. All reviewer comments are in black plain font, whereas our response is described in red plain font.

We have made the changes and corrections on the basis of the reviewer’s suggestions. We evaluated the comments and prepared a point-by-point response to each one of them.

Comments from the editors and reviewers:

The manuscript has been modified including most of the suggestions. Then, it can be now accepted for publication.

Reviewers' comments:

Reviewer 4

This study evaluated the use of four LAB starter cultures, previously selected for their properties as bioprotective agents, for sea bass and sea bream burgers biopreservation for storage at 4 °C for 10 days and at 8°C for the remaining 20 days in modified atmosphere packaging. Microbial populations, biochemical parameters (pH, TVB-N), and sensory properties of fish burgers were analyzed.

The work is interesting, but the presentation of the results and the discussion should be reviewed. Below is a list of suggestions.

Title: could be shorter

Thanks – Answer -Lines 2-3 - I changed it with: Improving the shelf-life of fish burgers made with a mix of sea bass and sea bream meat by bioprotective cultures

Line 19-20: This part of the sentence is not understood “followed by 30 days and volatilization for 18 days ….”

Thanks – Answer -Lines 18 - I eliminate the sentence : “followed by 30 days and volatilization for 18 days ….”

Line 20-22: This sentence contradicts what was said above: “In general, microbial and physicochemical parameters were not affected by the use of starter cultures but by the storage time, except for TVB-N values, which were significantly higher in the uninoculated burgers”

Thanks – Answer –  Lines 19-23 - I changed sentence -  Starter cultures impacted the microbial populations, biochemical parameters (pH, TVB-N), and sensory properties of fish burgers, during 10 days of storage at 4 °C and then 20 days at 8 °C in modified atmosphere packaging (MAP). Also storage time influenced the microbial and physicochemical characteristics of all the tested samples, except for TVB-N values, which were significantly higher in the uninoculated burgers.

Line 42: This quote cannot be found: [www.mangimialimenti.it]

Thanks – Answer – Lines 42 - Now this link is closed – I changed it : www.repubblica.it, www.ismeamercati.it.

Line 82: Clarify what the acronym VP means.

Thanks – Answer – LINES 80 - the acronym is vacuum packaging, I add under vacuum 

Line 104-107: The objective is not worded properly.

Thanks – Answer – Lines 97-98 - The aim of the present study was to develop fish burgers made with a mix of sea bass and sea bream meat and to improve their shelf life by bioprotective cultures.

Line 136-137: Clarify what the acronyms means.

Thanks – Answer – Lines 127-132 - The acronyms indicate the composition of the packaging : PET (Polyethylenterephtalat/PE (Polyethylene)/EVOH (Ethylene vinyl alcohol)/PE (Polyethylene) (ANTIFOG – EVOH (Ethylene vinyl alcohol). The trays were laminated with a top film consisting of APET Amorphous Polyethylenterephtalat/PE (Polyethylene)/EVOH (Ethylene vinyl alcohol)/PE (Polyethylene).

Line 137-138: Why was a change in storage conditions made to day 10?

Thanks – Answer – Lines 133-138 - I specified: According to the challenge test proposed by AFNOR NF V01-003 – 2004 - Hygiene and safety of foodstuffs - Guidelines for the design of an ageing test protocol for the validation of a microbiological lifetime, which reports for chilled perishable goods that in case of the cold chain not sufficiently guaranteed, it must be used two temperature:  1/3 of the shelf life at T1 (4 °C) and 2/3 at T2 (8 °C – abuse temperatures).

Line 149-150: Incubation of MRS agar: up to 3 days at 35ºC or up to 5 days at 30ºC.

Thanks – Answer – Lines 150 -  I correct up to 5 days at 30 °C

Line 152-154: The detection methodology for L. monocytogenes and Salmonella spp. should be briefly described.

Thanks – Answer  - Lines 153-163 - I add: Listeria monocytogenes – (Briefly – 25 g product were added to 225 ml of Fraser broth (Oxoid, Italy) incubated at 30 °C for 24 h, then an aliquot of this broths was streaked on Chromocult Listeria Agar according to Ottaviani/Agosti agar (Biolife, Italy) incubated at 37 °C for 24 h. On this agar L. monocytogenes produce typical blue-green colonies surrounded by an opaque halo);

Salmonella (briefly: 25 g product were added to 225 ml of Buffered Peptone Water (BPW, Oxoid, Italy) incubated 18 h at 37 °C, then 1 ml of BPW in 9 ml of Rappaport Vassiliadis broth (RVB, Oxoid, Italy) incubated at 42 °C for 18-24 h. An aliquot of RVB was streaked on Xylose Lysine Tergitol 4 agar (Oxoid, Italy) incubated at 37 °C for 24 h. On this agar the black or center black colonies were presumptive Salmonella.

Line 154-156: The identification methodology of MRS and TSM isolates should be briefly described.

Thanks – Answer  - Lines 165-169 - From MRS and TSM agars 5 colonies per plate were isolated and previous purification were subjected to Polymerase chain reaction (PCR) and the PCR products, after purification, were sent to a commercial facility for sequencing (MWG Biotech, Ebersberg, Germany). The sequences were aligned in GenBank using the Blast program version 2.2.18. .

Line 160-162: The methodology to determine TVB-N and oxidative stability should be briefly described.

Thanks – Answer – Lines 176-186 - Briefly: the total volatile basic nitrogen (TVB-N) was estimated by boiling a mix of distilled water (50 mL) and 10 g product in presence of MgO (25 mL, 2% w/v). The distillate was collected in a solution of boric acid and titrated with sulfuric acid in the presence of methyl red. Data are expressed in mg Nitrigen/100g. Thiobarbituric acid value (TBARS) was determined directly by spectrophotometric quantification of compounds obtained by the distillation of a mix consisting of distilled water (50 mL) and fish product (10 g), acidified with hydrochloric acid (2.5 mL, 4 N) until pH 1.5. Then, 5 mL of the distillate was treated with 5 mL of a solution of thiobarbituric acid (TBA), obtained by mixing TBA (0.2883) in acetic acid (90%) placed in boiled water for 35 min. After cooling, the solution was read at 538 nm. Three analyses were performed at each sampling point and data are expressed in nmol malonaldehyde/g.

.

Line 199-200: “Total bacterial counts were similar for all the samples 199 throughout the storage period and were not affected by starter inoculum.” On day 0 and 12 there are differences

Thanks – Answer – you are right – I changed – Lines 225-227 - In all samples CBT increased till 12 days storage, reaching values between 5.32 and 6.47 Log CFU/g, then it decreased and at the end it was between  3.01 and 3.58 Log CFU/g.

Line 214-216: “In fact, the pH of the control samples was 5.56, which was significantly higher (p < 0.05) than that of the inoculated samples, which had pH values lower than 5.” Sample F-106 presented a pH value greater than 5.

Thanks, Answer – you are right: I correct : Lines 243-244 - which was significantly higher (p < 0.05) than that of the inoculated samples, which had pH values between 4.67 and 5.09.

Line 263-264: “Alcohols, and in particular 1-propanol, isopropyl alcohol and 1-penten-3-ol, were detected in higher amounts in the control at time 0” Time 0 data is missing for inoculated samples in the table.

Thanks – Answer – The control at time 0 represent the time 0 of the treated samples.

Line 268- 270. “After 6 days, aldehydes showed a small decrease in the control and in the fish burgers 268 containing the culture F-106, while an increase was observed when BOX-57, FP-50 and 269 LAK-23 were added” The way in which the data is presented does not speed up the reading of the evolution of the data over time.

Thanks – Answer – Lines 287-289 - I explain well : After 6 days, aldehydes showed a small decrease in the control and in the fish burgers containing the culture F-106, while an increase was observed in BOX-57, FP-50 and LAK-23 samples.

Line 301: “Factor 1 explains 27.43% and Factor 301 2 explains 23.45% of the variability.” The definition of factor 1 factor 2 is missing.

Thanks – Answer: The principal component analysis is based on the definition of new derived variable formed as a linear combination of the original variables that explains the most variance of the original data. The process consists in the definition of a number of new variables which number is n-1 where n is the number of the original variables (in this case volatile compounds) taken into consideration. Factor 1 and Factor 2 represent the first two new variables, i.e. the two variables explaining the greater variability (in our case, approx 50% of the total variability). In figure it is also reported the contribution of each original variables to the new variables 1 and 2 obtained. As an example, pentanal contributed for 0.0385 (3.85%) to Factor 1 and 0.0821 (8.21%) to Factor 2. From another point of view, butanoic acid (0.0852) is the major contributor to Factor 1 while 1-penten-3-ol is the major contributor to Factor 2 (0.0918)

Line 385: “20 CFU/g”. Is it correct?

Thanks – Answer – Lines 424 – Yes, it is correct.

Line 398-340: “… the starter cultures added to the burgers, 398 which had homofermentative metabolism, prevailed over the indigenous LAB and re- 399 stricted the species variability…” Is this conclusion correct? Do the data obtained allow us to reach this statement? Table 1: Not all data corresponds to Log.

Thanks – Answer – Lines 436-440 Yes, it is correct – I add to table 1 the expressed value : Microorganisms – log CFU/g; TVB-N – mg N/100g (see the table in the revised paper).

Line 401: It is not correct to quote figures in the discussion

Thanks – Answer – I eliminate it from discussion.

Line 427-429: “It can be concluded that the bioprotective cultures used in this study resulted in effective elimination of bloating spoilage” Is it correct?

Thanks – Answer – Lines 473-476 - It is correct, but the part is eliminated and consider Lines 436-440.

Line 430: The figure should be cited in the results section.

Thanks – Answer – Lines 476 - I eliminate it.

Line 461: “The strain Lacticaseibacillus casei (F-106) showed slower kinetics” Is it correct?

Thanks – Answer – Line 496 -  it is correct.

Line 470: “Inoculated burgers did not present odors, flavors or sticky white slime indicative of deterioration.” Is this described in the results?

Thanks – Answer – Lines 366-368 - No it is not described in the results. I add Finally inoculated burgers did not present odors, flavors or sticky white slime indicative of deterioration. In contrast, a sticky-white slime was observed in some uninoculated burgers.

Table 2: The data is outside the corresponding line. Some of the acronyms are not clarified in the legend. It does not present the SD. Nor is it discriminated which results are statistically different. This table should be redesigned to provide easy reading of the results.

Thanks – Answer – Lines 298-401 - Table 2 has been modified to facilitate comprehension and the caption has been rewritten to better clarify the acronyms of the different burgers. In addition, the presence of statistical differences for each volatile compound between samples collected at each storage time has been indicated by adding different letters. See the table in the revised paper.

Table 3: It is not understood what the proportions indicated in the table refer to and how the final score is calculated. In materials and methods it is not explained.

Thanks – Answer – Lines 205-213 - I try to explain well:

The sensory evaluation panel consisted of 12 nonprofessional assessors. The cooked burgers were presented on white plates at room temperature. Assessors were asked to evaluate the following descriptors: odor (fermentation, rancid, fishy), taste (sweet, sour, pungent, rancid), flavor (ammonia, sweet, sour, bitter) and appearance (slime). The 12 assessors evaluated the presence or the absence of each of the nine descriptors. The results stated for each sample is the sum of the assessors who considered the presence of the descriptor out of the total of the assessors [36,37]. Then the final score is calculated asking to the panelists to give a general evaluation of sensory quality of the products within the scale from 1 (excellent) to 4 (worst).

Lines 370-374 - Legend: CTRL: no starter; BOX-57 (C. maltaromaticum, C. divergens, L. sakei), F-106 (Lacticaseibacillus casei), FP-50 (C. maltaromaticum, C. divergens); LAK-23 (L. sakei). Data: sum of the assessors identifiers of the presence of the descriptor out of the total of the assessors; a Final scores: the assessors requested to ranked the products within the scale from 1 (excellent) to 4 (worst).

Reviewer 5 Report

This manuscript reports an experiment on “Development of fish burgers made with a mix of sea bass and sea bream meat and added with bioprotective cultures to determine the shelf life during refrigerated storage”. This is an interesting area of work. The manuscript is well written, where the experiment is well designed and discussed. However, there is one main issue with the manuscript where the unit expression is not available in material and method section for of microbiological and physio-analysis. The unit is also not written in Table 1. The manuscript needs some minor corrections before the manuscript can be considered as an accepted manuscript.

 Points for the authors to address:

1.      Section 2.4 ….. how did TBRAS measure in this experiment, ? Please provide more information of this method.

2.      Section 2.6 … … line 184 ….. from the best to the worst ……  how did each parameter assess based on this description? More details information is required for section 2.6. sensory analysis.

Author Response

Response to Reviewer 5 Comments

Dear Referee

Manuscript ID: microorganisms-1884900

Improving the shelf-life of fish burgers made with sea bass and sea bream meat by bioprotective cultures.

Journal: Microorganisms

Answer to the referee.

The authors would like to thank the reviewers for their careful reading of the manuscript and the resulting constructive comments and suggestions. Basically, we agree with all of the points raised by the reviewers, and wherever possible the manuscript has been modified as recommended. All reviewer comments are in black plain font, whereas our response is described in red plain font.

We have made the changes and corrections on the basis of the reviewer’s suggestions. We evaluated the comments and prepared a point-by-point response to each one of them.

Comments from the editors and reviewers:

The manuscript has been modified including most of the suggestions. Then, it can be now accepted for publication.

Reviewers' comments:

Reviewer 5

This manuscript reports an experiment on “Development of fish burgers made with a mix of sea bass and sea bream meat and added with bioprotective cultures to determine the shelf life during refrigerated storage”. This is an interesting area of work. The manuscript is well written, where the experiment is well designed and discussed. However, there is one main issue with the manuscript where the unit expression is not available in material and method section for of microbiological and physio-analysis. The unit is also not written in Table 1. The manuscript needs some minor corrections before the manuscript can be considered as an accepted manuscript.

I add the unit either for microorganisms (lines 152-153 ) or physico-chemical analysis 176-186

Points for the authors to address:

  1. Section 2.4 ….. how did TBRAS measure in this experiment, ? Please provide more information of this method.

Thanks – Answer – Lines 176-186 - Briefly: the total volatile basic nitrogen (TVB-N) was estimated by boiling a mix of distilled water (50 mL) and 10 g product in presence of MgO (25 mL, 2% w/v). The distillate was collected in a solution of boric acid and titrated with sulfuric acid in the presence of methyl red. Data are expressed in mg Nitrigen/100g. Thiobarbituric acid value (TBARS) was determined directly by spectrophotometric quantification of compounds obtained by the distillation of a mix consisting of distilled water (50 mL) and fish product (10 g), acidified with hydrochloric acid (2.5 mL, 4 N) until pH 1.5. Then, 5 mL of the distillate was treated with 5 mL of a solution of thiobarbituric acid (TBA), obtained by mixing TBA (0.2883) in acetic acid (90%) placed in boiled water for 35 min. After cooling, the solution was read at 538 nm. Three analyses were performed at each sampling point and data are expressed in nmol malonaldehyde/g.

  1. Section 2.6 … … line 184 ….. from the best to the worst ……  how did each parameter assess based on this description? More details information is required for section 2.6. sensory analysis.

Thanks – Answer – Lines 205-213 - I try to explain well:

The sensory evaluation panel consisted of 12 nonprofessional assessors. The cooked burgers were presented on white plates at room temperature. Assessors were asked to evaluate the following descriptors: odor (fermentation, rancid, fishy), taste (sweet, sour, pungent, rancid), flavor (ammonia, sweet, sour, bitter) and appearance (slime). The 12 assessors evaluated the presence or the absence of each of the nine descriptors. The results stated for each sample is the sum of the assessors who considered the presence of the descriptor out of the total of the assessors [36,37]. Then the final score is calculated asking to the panelists to give a general evaluation of sensory quality of the products within the scale from 1 (excellent) to 4 (worst).

Lines 370-374 - Legend: CTRL: no starter; BOX-57 (C. maltaromaticum, C. divergens, L. sakei), F-106 (Lacticaseibacillus casei), FP-50 (C. maltaromaticum, C. divergens); LAK-23 (L. sakei). Data: sum of the assessors identifiers of the presence of the descriptor out of the total of the assessors; a Final scores: the assessors requested to ranked the products within the scale from 1 (excellent) to 4 (worst).

Round 2

Reviewer 1 Report

(1)References can be limited to less than 60;

(2)Page 2 line62-62, this should be belonged to the same paragraph.

(3) Page 2 line 80 Under Vacuum? under vacuum?

(4) Page 4 line 190,error.

(5)p < 0.05. “p”Italics

Author Response

Response to Reviewer 1 Comments

Dear Referee

Manuscript ID: microorganisms-1884900

  

Improving the shelf-life of fish burgers made with sea bass and sea bream meat by bioprotective cultures.

Journal: Microorganisms

Answer to the referee.

The authors would like to thank the reviewers for their careful reading of the manuscript and the resulting constructive comments and suggestions. Basically, we agree with all of the points raised by the reviewers, and wherever possible the manuscript has been modified as recommended. All reviewer comments are in black plain font, whereas our response is described in red plain font.

We have made the changes and corrections on the basis of the reviewer’s suggestions. We evaluated the comments and prepared a point-by-point response to each one of them.

Comments from the editors and reviewers:

The manuscript has been modified including most of the suggestions. Then, it can be now accepted for publication.

Reviewers' comments:

Reviewer 1  

References can be limited to less than 60;

Thanks – Answer The high number of references needs to better explain the problem and the discussion. We think they are necessary.

(2)Page 2 line62-62, this should be belonged to the same paragraph.

Thanks – Answer Lines 59 - I correct it

(3) Page 2 line 80 Under Vacuum? under vacuum?

Thanks – Answer Line 76 I correct it

(4) Page 4 line 190,error.

Thanks – Answer Line 185 I correct it

(5)p < 0.05. “p”Italics

Thanks – Answer Lines 231 – 239 - 248 – 261  I made it.

Reviewer 4 Report

Some minor modifications need to be done:

Line 19-21: This sentence is repeated: “Starter cultures impacted the microbial populations, biochemical parameters (pH, TVB-N), and sensory properties of fish burgers, during 10 days of storage at 4 °C and then days at 8 °C in modified atmosphere packaging (MAP).”

Line 204: Sensory analyses: It must be specified which samples and their quantity were evaluated.

Line 373-374: “Final scores: the assessors requested to ranked the products within the scale from 1 (excellent) to 4 (worst)”:  in the table one of the scores is 5.

Author Response

Response to Reviewer 4 Comments

Dear Referee

Manuscript ID: microorganisms-1884900

  

Improving the shelf-life of fish burgers made with sea bass and sea bream meat by bioprotective cultures.

Journal: Microorganisms

Answer to the referee.

The authors would like to thank the reviewers for their careful reading of the manuscript and the resulting constructive comments and suggestions. Basically, we agree with all of the points raised by the reviewers, and wherever possible the manuscript has been modified as recommended. All reviewer comments are in black plain font, whereas our response is described in red plain font.

We have made the changes and corrections on the basis of the reviewer’s suggestions. We evaluated the comments and prepared a point-by-point response to each one of them.

Comments from the editors and reviewers:

The manuscript has been modified including most of the suggestions. Then, it can be now accepted for publication.

Reviewers' comments:

Reviewer 4

Some minor modifications need to be done:

Line 19-21: This sentence is repeated: “Starter cultures impacted the microbial populations, biochemical parameters (pH, TVB-N), and sensory properties of fish burgers, during 10 days of storage at 4 °C and then days at 8 °C in modified atmosphere packaging (MAP).”

Thanks – Answer Lines 16-19 I eliminated the repetitions 

Line 204: Sensory analyses: It must be specified which samples and their quantity were evaluated

Thanks – Answer I add Line 201-202 - Ten burgers of the control and of each treatment were evaluated

Line 373-374: “Final scores: the assessors requested to ranked the products within the scale from 1 (excellent) to 4 (worst)”:  in the table one of the scores is 5.

Thanks – Answer Line 209 I correct it